JCB Journal of Cell Biology

# KPNA3 regulates histone locus body formation by modulating condensation and nuclear import of NPAT

Shui Bo Xu[1,2,3]*, Xiu Kui Gao[4]*, Hao Di Liang[1], Xiao Xia Cong[1,2], Xu Qi Chen[1], Wen Kai Zou[1,2], Jia Li Tao[1,2], Zhao Yuan Pan[1], Jiao Zhao[5], Man Huang[1,3], Zhang Bao[6], Yi Ting Zhou[2,3,7], and Li Ling Zheng[1,3,7]

The histone locus body (HLB) is a membraneless organelle that determines the transcription of replication-dependent histones. However, the mechanisms underlying the appropriate formation of the HLB in the nucleus but not in the cytoplasm remain unknown. HLB formation is dependent on the scaffold protein NPAT. We identify KPNA3 as a specific importin that drives the nuclear import of NPAT by binding to the nuclear localization signal (NLS) sequence. NPAT undergoes phase separation, which is inhibited by KPNA3-mediated impairment of self-association. In this, a C-terminal self-interaction facilitator (C-SIF) motif, proximal to the NLS, binds the middle 431–1,030 sequence to mediate the self-association of NPAT. Mechanistically, the anchoring of KPNA3 to the NPAT-NLS sterically blocks C-SIF motif-dependent NPAT self-association. This leads to the suppression of aberrant NPAT condensation in the cytoplasm. Collectively, our study reveals a previously unappreciated role of KPNA3 in modulating HLB formation and delineates a steric hindrance mechanism that prevents inappropriate cytoplasmic NPAT condensation.

## Introduction

In addition to canonical organelles surrounded by the lipid bilayer membrane, eukaryotic cells also contain many membraneless compartments formed by phase separation or biomolecular condensation (Alberti and Hyman, 2021; Dall'Agnese et al., 2022; Hirose et al., 2023; Ladbury et al., 2023; Lin et al., 2023; Lyon et al., 2021; Seydoux et al., 2023). These membraneless condensates include stress granules and signalosomes in the cytoplasm, as well as numerous nuclear bodies including promyelocytic leukaemia (PML) nuclear bodies, Cajal bodies, and histone locus bodies (HLB) (Bhat et al., 2021; Dall'Agnese et al., 2022; Gao et al., 2022; Gao et al., 2023; Kang et al., 2022; Noda et al., 2020; Nong et al., 2021; Parchure et al., 2022; Sabari et al., 2020). Among these nuclear bodies, the HLBs are specialized nuclear structures for the transcription and processing of replication-dependent (RD) histone mRNAs (Duronio and Marzluff, 2017; Geisler et al., 2023; Kemp et al., 2021; Nizami et al., 2010; Tatomer et al., 2016; White et al., 2007). During cell proliferation, both RD-histones and DNA

need to be concurrently duplicated for the packaging of newly replicated chromosomes (Armstrong et al., 2023; Armstrong and Spencer, 2021; Marzluff and Koreski, 2017; Mendiratta et al., 2019). Thus, the assembly of HLBs is essential for cell proliferation and must be tightly controlled. However, any mechanism to ensure that HLBs form only in nuclei and not in the cytoplasm has remained elusive.

Nuclear protein at the ataxia-telangiectasia locus (NPAT), which is localized primarily to the nucleus, plays essential roles in regulating the transcription of RD-histones (Cui et al., 2022; Ma et al., 2000; Wei et al., 2003; Ye et al., 2003; Zhao et al., 2000; Bruhn et al., 2020). As a scaffold protein, NPAT recruits other HLB components such as FLICE-associated huge protein (FLASH) and U7 small nuclear ribonucleoprotein (U7 snRNP) to initiate HLB formation (Duronio and Marzluff, 2017; Nizami et al., 2010; Tatomer et al., 2016; Terzo et al., 2015; White et al., 2007; Yang et al., 2014). The dysregulation or mutation of NPAT is related to cancer susceptibility (Ishikawa et al., 2011;

....................................................................................................................................................................................................................

[1]Department of General Intensive Care Unit and Department of Biochemistry of the Second Affiliated Hospital, Liangzhu Laboratory, Zhejiang University School of Medicine, Hangzhou, China; [2]ZJU-UoE Institute, Dr. Li Dak Sum & Yip Yio Chin Center for Stem Cell and Regenerative Medicine, Zhejiang University School of Medicine, Hangzhou, China; [3]Key Laboratory of Multiple Organ Failure (Zhejiang University), Ministry of Education, Hangzhou, China; [4]International Institutes of Medicine, the Fourth Affiliated Hospital of Zhejiang University School of Medicine, Yiwu, China; [5]Department of Endocrinology, Hangzhou First People's Hospital, Hangzhou, China; [6]Department of Respiratory Medicine, the First Affiliated Hospital, Zhejiang University School of Medicine, Hangzhou, China; [7]Department of Orthopaedic Surgery of the Second Affiliated Hospital, Zhejiang University School of Medicine, Hangzhou, China.

*S.B. Xu and X.K. Gao contributed equally to this work. Correspondence to Li Ling Zheng: zhengliling@zju.edu.cn; Yi Ting Zhou: zhouyt@zju.edu.cn; Zhang Bao: baozhang@zju.edu.cn.

Kalla et al., 2007; Milne et al., 2017; Saarinen et al., 2011; Wright et al., 2017; Terradas et al., 2024). Our and other's previous studies have shown that HiNF-P and Cpn10/Hsp10 interact with and stabilize NPAT and that NPAT is modulated by CDK/Cyclin E-mediated phosphorylation during the cell cycle (Miele et al., 2005; Zheng et al., 2015). Notably, the condensation of Mxc, the *Drosophila* NPAT homolog, has been shown to drive the formation of HLBs (Hur et al., 2020; Koreski et al., 2020). Yet, the formation of HLBs only occurs in nuclei. This raises a question relating to the mechanism that prevents the aberrant NPAT condensation and HLB formation in the cytoplasm. Notably, the mechanism underlying the nuclear import of NPAT has also remained elusive.

The transport of nuclear localization signal (NLS)-containing protein from the cytoplasm into the nucleus is mediated by the karyopherin (also referred to as importin) superfamily members, which are further categorized into karyopherin α (KPNAs) and β (KPNBs), based on their functional features (Paci et al., 2021; Wing et al., 2022). KPNAs act as adaptor molecules between the NLS-bearing cargo proteins and the KPNB1 (importin β1) carrier molecule in the cytoplasm (Miyamoto et al., 2016; Pumroy and Cingolani, 2015). Notably, karyopherins also play roles in regulating phase separation of RNA-binding proteins (Guo et al., 2018; Hofweber et al., 2018; Springhower et al., 2020; Yoshizawa et al., 2018) and preventing nucleocytoplasmic leakage (Kalita et al., 2022). In this study, we revealed that NPAT specifically interacts with KPNA3 by its C-terminal NLS. We further verified that KPNA3 determines the nuclear import of NPAT and the formation of the HLB. Importantly, the anchoring of KPNA3 to the NPAT-NLS sterically blocks the self-association of NPAT. This leads to the suppression of NPAT condensation. Our findings indicate that the KPNAs-mediated decondensation may serve as a common strategy to regulate the formation of nuclear organelles.

## Results

### NPAT specifically interacts with KPNA3

KPNAs perform the indispensable role of ferrying proteins from the cytoplasm to the nucleus (Miyamoto et al., 2016; Pumroy and Cingolani, 2015). Administration of a KPNA-specific inhibitor, ivermectin (25 μM), that interferes with importin α/β-dependent nuclear import (Wing et al., 2022), dramatically reduced the HLB numbers in MCF-7 cells (Fig. 1 A). MCF-7 breast cancer cell line was used for immunofluorescence study because of its ease of transfection and the association of NPAT with breast cancer (Milne et al., 2017; Rogers et al., 2015). We also measured the expression levels of RD-histones from the major histone gene locus *HIST1* and a minor histone gene locus *HIST2* (Marzluff et al., 2002). Indeed, ivermectin treatment downregulated the expression levels of RD-histone mRNA (Fig. 1 B). NPAT acts as the key regulator for HLB formation and maintenance in mammalian cells (Duronio and Marzluff, 2017; Nizami et al., 2010; Tatomer et al., 2016; White et al., 2007). Here, we established a stable cell line (293T-GFP-NPAT cell), which expressed GFP-NPAT in an inducible manner. By titrating the concentration and the induction time of the inducer, doxycycline

(DOX), we could express exogenous GFP-NPAT protein at close to endogenous levels in parental cells (Fig. S1 A). GFP-NPAT was noted to display foci structures (Fig. S1 B), which colocalized with FLASH, another maker for HLB, in the 293T-GFP-NPAT stable cells (Fig. S1 C). The foci formation of GFP-NPAT was then disrupted upon ivermectin treatment (Fig. 1 C). All these data indicate the involvement of KPNA in regulating the nuclear import of NPAT.

To gain mechanistic insights into NPAT nuclear localization, we screened for the KPNA(s) that may be responsible for the transportation of NPAT into the nucleus. Coimmunoprecipitation (Co-IP) assays demonstrated that NPAT specifically interacts with the KPNA3/importin α4 (Fig. 1 D), but not other KPNA family members. 293T cell line was used because of its ease of transfection and production of recombinant proteins (Stepanenko and Dmitrenko, 2015). We further prepared three NPAT truncation mutants that encompass the N-terminal 1–430 sequence (N-region), the middle 431–1,030 sequence (M-region), and the C-terminal 1,031–1,427 sequence (C-region) to distinguish the specific region of NPAT interaction with KPNA3 (Fig. 1 E). Co-IP assay results showed that KPNA3 is mainly bound to the C-terminal 1,031–1,427 sequence of NPAT (Fig. 1 F), which contains a previously reported NLS (1,368–1,405) (Sagara et al., 2002). The ectopically expressed GFP-NPAT-C-region displayed nuclear localization (Fig. 1 H). In contrast, an NPAT-C-region ΔNLS mutant (Fig. 1 G) demonstrated cytosolic localization (Fig. 1 H). Consistent with the notion that NLS mediates the interaction between KPNAs and cargoes (Miyamoto et al., 2016; Pumroy and Cingolani, 2015), the NPAT-C-region ΔNLS mutant failed to bind KPNA3 (Fig. 1 I). These findings indicate that KPNA3 is a novel specific interacting partner of NPAT and a prime candidate for involvement in regulating the nuclear import of NPAT.

### KPNA3 mediates the nuclear import of NPAT

To test the hypothesis that KPNA3 mediates the nuclear localization of NPAT, we first assessed the subcellular distribution of endogenous NPAT in control or siRNA-induced KPNA3 knockdown cells. Subcellular fractionation analysis revealed decreased endogenous nuclear NPAT levels in the KPNA3-depleted cells (Fig. 2 A and Fig. S2 A). Likewise, ivermectin treatment (25 μM) also reduced the expression levels of nuclear NPAT (Fig. S2 B). Treating cells with a higher concentration of ivermectin (50 μM) resulted in aberrant cytoplasmic NPAT foci formation (Fig. S2 C). We then further examined the roles of KPNA3 in regulating HLB and RD-histone transcription. Indeed, the knockdown of KPNA3 reduced the number of NPAT-positive HLB foci in the nuclei (Fig. 2 B and Fig. S2 D). In 293T-GFP-NPAT cells, knockdown of KPNA3 consistently led to a reduced nuclear fraction of GFP-NPAT and impairment of GFP-NPAT foci formation (Fig. 2, C and D). Since siRNA-mediated knockdown for 48 h only partially reduced the KPNA3 expression (Fig. S2 D), we further performed a time-extended knockdown of KPNA3 (72 h), leading to a mild effect of aberrant formation of NPAT foci in the cytoplasm (Fig. S2 E). To verify the roles of KPNA3 in regulating NPAT foci formation, we further performed ivermectin treatment (25 μM) in cells transfected with KPNA3 siRNA. The

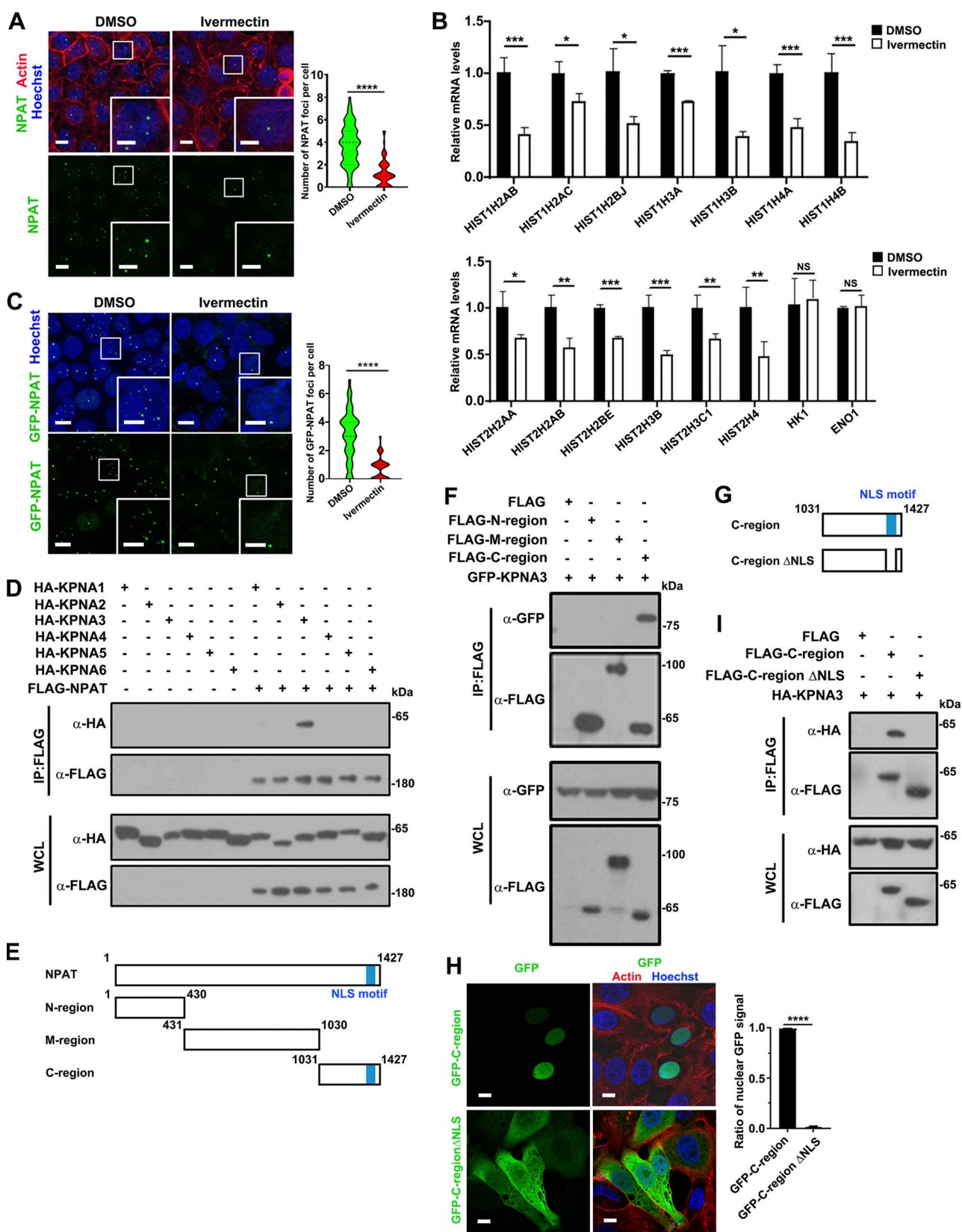

Figure 1. **KPNA3 interacts with the NLS of NPAT. (A)** MCF-7 cells treated with or without ivermectin (25 µM) were stained with phalloidin for actin and NPAT antibody for HLB. Bar: 10 µm. The insets show the magnification of the boxed region (scale bar, 5 µm). The numbers of HLB foci per cell were quantified as violin plots (DMSO: *n* = 126; ivermectin: *n* = 132). ****: P < 0.0001. **(B)** MCF-7 cells treated with or without ivermectin (25 µM) were subjected to qPCR analysis for transcription levels of the indicated RD histone genes and two control genes. Data in the bar graphs represent the means ± SEM values of the relative transcription levels for three independent experiments. *: P < 0.05, **: P < 0.01, ***: P < 0.001, ****: P < 0.0001, ns: not significant. **(C)** Confocal

images of 293T-GFP-NPAT cells treated with or without ivermectin (25 µM). Bar: 10 µm. The insets show the magnification of the boxed region (scale bar, 5 µm). The GFP-NPAT foci per cell were quantified as violin plots (DMSO: n = 85; ivermectin: n = 73). ****: P < 0.0001. **(D)** Coimmunoprecipitation of FLAG-NPAT and HA-tagged KPNA family members. **(E)** Schematic diagram of NPAT and its truncation mutants. **(F)** Coimmunoprecipitation of FLAG-tagged NPAT truncation mutants and GFP-KPNA3. **(G)** Schematic diagram of the NPAT C-region and its ∆NLS deletion mutant (∆1,368–1,405). **(H)** Confocal images of representative MCF-7 cells expressing the GFP-NPAT C-region and its NLS deletion mutant. Scale bar: 5 µm. Data in the bar graphs represent the means ± SD values of the ratio of nuclear GFP signal of WT or NLS deletion mutant for three independent experiments. ****: P < 0.0001. **(I)** Coimmunoprecipitation of the FLAG-tagged NPAT C-region or its ∆NLS deletion mutant and HA-KPNA3. Source data are available for this figure: SourceData F1.

combination of KPNA3 inhibition and depletion not only caused a more significant reduction of the NPAT foci in nuclei but also led to more obvious aberrant NPAT foci formation in the cytoplasm (Fig. 2 E). This finding indicates that KPNA imports NPAT into nuclei and prevents aberrant NPAT condensation in the cytoplasm.

Depletion of KPNA3 reduced the mRNA levels of RD-histones from both loci (Fig. 2 F). In line with this, both colony formation and CCK8 assays demonstrated that KPNA3 knockdown inhibited cell proliferation (Fig. 2, G and H). We thus concluded that KPNA3 determined the nuclear import of NPAT to regulate HLB formation and RD-histone transcription (Fig. 2 I).

## NPAT undergoes phase separation/condensation

It has been recently found that the multi-sex combs (Mxc) protein, the *Drosophila* ortholog of NPAT, undergoes phase separation to drive the formation of the HLB (Hur et al., 2020). We thus considered that NPAT may also undergo condensation. To investigate this, we first examined GFP-NPAT in cells since GFP could act as a tag for performing fluorescence recovery after the photobleaching (FRAP) and foci fusion analysis, which permits examination of the liquid property of NPAT foci. Sequence analysis using the PONDR program showed major parts of NPAT to be intrinsically disordered regions (IDRs) (Fig. 3 A). Indeed, the GFP-NPAT formed spherical structures in cells, which could undergo fusion (Fig. 3 B; and Fig. S3, A and B). FRAP of GFP-NPAT droplets also revealed a fast recovery rate (Fig. 3 C). These data indicated the liquid property of GFP-NPAT droplets, which are highly dynamic and readily exchange molecules with the surrounding cytosol.

We next purified recombinant FLAG-NPAT to perform an in vitro phase separation assay by differential interference contrast (DIC) microscopy analysis. Our measurement displayed that the endogenous concentration of NPAT protein in cells was ~10–33 µM (10.4 µM in MCF-7 and 33.1 µM in 293T) (Fig. S3 C). Polyethylenglycol (PEG) was used to mimic the crowding conditions in cells. Formation of NPAT droplets was observed at a concentration of 6.25 µM and increased protein concentrations led to larger sizes and increased numbers of NPAT spheres (Fig. 3 D). Elevation of the PEG concentration significantly promoted the formation of NPAT droplets (Fig. 3 E). By contrast, the addition of 1,6-hexanediol, an aliphatic alcohol that weakens hydrophobic interactions, reduced NPAT droplet formation (Fig. 3 F). Increasing salinity had no observable effect on the assembly of NPAT spherical structures (Fig. 3 G). We thus concluded that NPAT phase separation is mainly driven by hydrophobic, but not electrostatic interactions. Moreover, the NPAT droplets could be observed at 25°C but were much reduced when incubated at 4°C (Fig. 3 H). We thus verified that NPAT also undergoes phase separation (Fig. 3 I).

## The binding to KPNA3 suppresses NPAT condensation

The finding that NPAT undergoes phase separation raised the question of how NPAT condensation is suppressed in the cytoplasm. Previous studies have demonstrated that KPNB/importin-β suppresses the phase separation/condensation of the RNA-binding protein FUS and TDP-43 (Guo et al., 2018; Hofweber et al., 2018; Springhower et al., 2020; Yoshizawa et al., 2018). Since HLB is only formed in nuclei, we hypothesized that KPNA3 might inhibit the NPAT condensate in the cytoplasm to ensure its passing through the nuclear pore. To examine the potential involvement of KPNA3 in negatively regulating NPAT condensation, we first performed an in vitro phase separation assay. Indeed, purified KPNA3 at a concentration of 25 µM, which could not undergo phase separation (Fig. S4), significantly suppressed NPAT condensate (Fig. 4 A). Self-association-mediated protein polymerization is understood to be important for driving phase separation/condensation (Bienz, 2014). Coexpressing KPNA3 indeed reduced the self-association of NPAT (Fig. 4 B). Moreover, the punctate structures of transiently expressed GFP-NPAT were abolished upon coexpression with KPNA3 (Fig. 4 C). Whilst ectopic expression of KPNA3 consistently promoted the nuclear import of endogenous NPAT (Fig. 4 D), it also suppressed the formation of NPAT-positive HLBs in cells (Fig. 4 E). Consistently, expressing of KPNA3 reduced the expression levels of RD-histones (Fig. 4 F).

To further validate if the interaction with KPNA3 ameliorates NPAT condensation, we set out to identify the NPAT-binding region of KPNA3. The KPNAs contain a series of armadillo (Arm) repeats which act as cargo NLS-binding sites (Miyamoto et al., 2016) and where ARMs 2–4 are known to act as the major NLS binding groove for NLS sequences (Fontes et al., 2003). To identify the critical region for the KPNA3/NPAT interaction, we prepared three truncation mutants as shown in Fig. 4 G. Co-IP assays revealed that ARM 3–5 (KPNA3 ARM3–5) mediates the association to the NPAT C sequence, thus identifying the NLS-containing region of NPAT (Fig. 4 H). To further validate this, a KPNA3 mutant (∆ARM3–5) lacking ARM 3–5 was created (Fig. 4 I). As expected, the KPNA3 ∆ARM3–5 mutant failed to interact with the C sequence of NPAT (Fig. 4 J), and the KPNA3 ∆ARM3–5 could not suppress NPAT condensation (Fig. 4 K). In line with this, the NPAT-∆NLS mutant (Fig. 4 L), which exhibits cytoplasmic retention since it could not interact with KPNA3, also displayed cytoplasmic foci (Fig. 4 M). These findings indicate that binding with KPNA3 suppresses NPAT condensation (Fig. 4 N).

We further performed two experiments to distinguish the effects of mislocalization due to impaired import from the impact of KPNA3 binding on NPAT condensation. KPNA5 does not interact with NPAT (Fig. 1 D) and cannot suppress HLB formation (Fig. 5 A). First, we created a KPNA5-3-hybrid mutant by

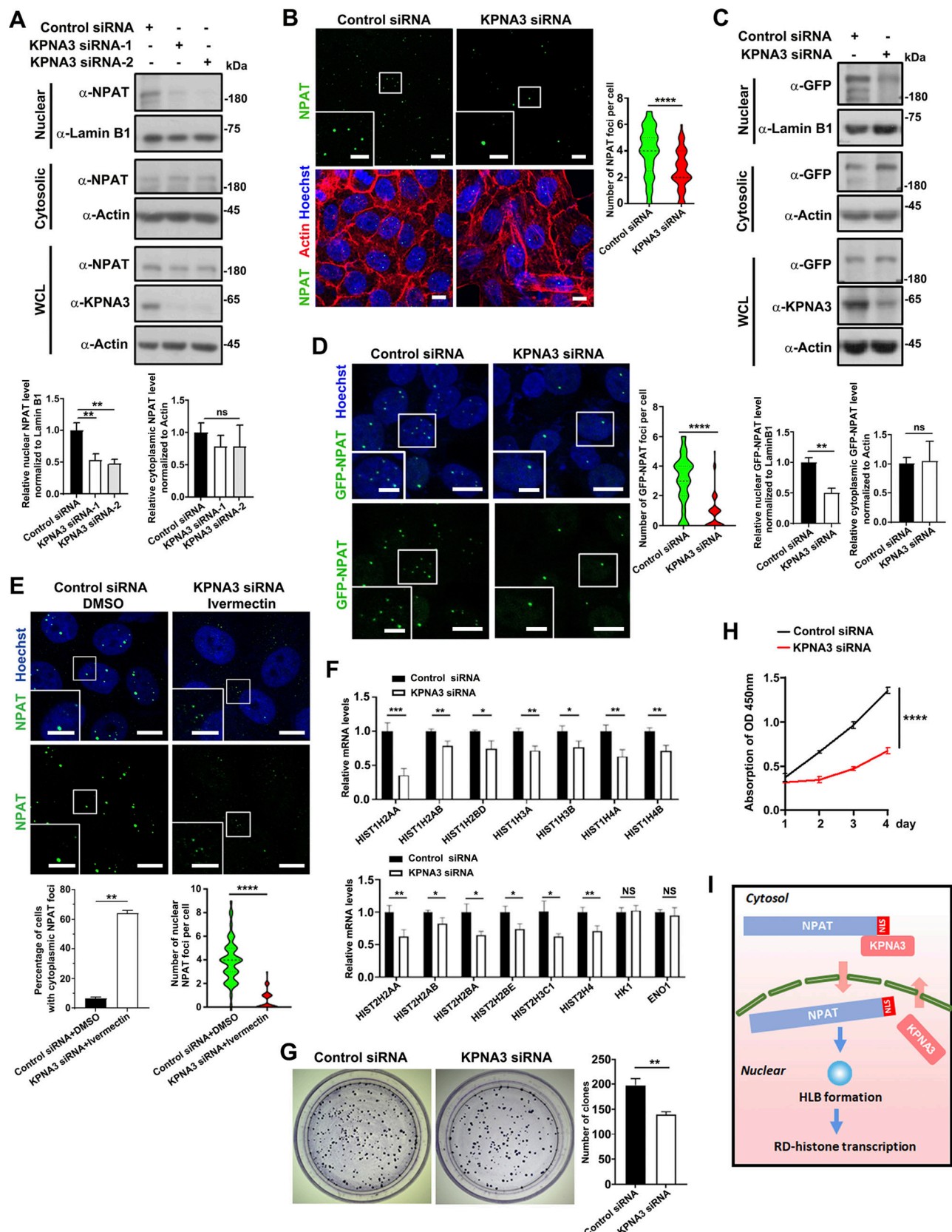

Figure 2. **KPNA3 determines the nuclear import of NPAT and HLB formation. (A)** Cytoplasmic and nuclear fractions were prepared from MCF-7 cells transfected with control or KPNA3 siRNA and then subjected to western blot analysis for examination of NPAT levels. Data in the bar graphs represent the means ± SEM values of the ratio of densities for three independent experiments. **: P < 0.01, ns: not significant. **(B)** Immunofluorescence staining of endogenous NPAT in MCF-7 cells transfected with control or KPNA3 siRNA for 48 h. Scale bar, 10 μm. The insets show the magnification of the boxed region

(scale bar, 5 µm). The endogenous NPAT foci per cell were quantified (control siRNA: n = 125; KPNA3 siRNA: n = 101). ****: P < 0.0001. **(C)** 293T cells stably expressing GFP-NPAT were transfected with control or KPNA3 siRNA. Cytoplasmic and nuclear fractions were prepared and subjected to western blot analysis for examination of GFP-NPAT levels. Data in the bar graphs represent the means ± SEM values of the ratio of densities for three independent experiments. **: P < 0.01, ns: not significant. **(D)** Confocal images of the 293T-GFP-NPAT cells transfected with control or KPNA3 siRNA. The insets show the magnification of the boxed region (scale bar, 5 µm). The GFP-NPAT foci per cell were quantified (control siRNA: n = 83; KPNA3 siRNA: n = 65). ****: P < 0.0001. **(E)** Immunofluorescence staining of endogenous NPAT in MCF-7 cells transfected with control siRNA or cells transfected with KPNA3 siRNA for 48 h and treated with ivermectin (25 µM). Scale bar, 10 µm. The insets show the magnification of the boxed region (scale bar, 5 µm). The ratio of cells with NPAT foci in cytoplasm was quantified and data in the bar graphs represent the means ± SD value for three independent experiments. **: P < 0.01. The nuclear NPAT foci per cell were quantified (control siRNA: n = 226; KPNA3 siRNA: n = 158). ****: P < 0.0001. **(F)** MCF-7 cells transfected with control or KPNA3 siRNA were subject to qPCR analysis for transcription levels of the indicated RD histone genes and two control genes. Data in the bar graphs represent the means ± SEM values of the relative transcription levels for three independent experiments. *: P < 0.05, **: P < 0.01, ***: P < 0.001, ****: P < 0.0001, ns: not significant. **(G)** Control or KPNA3 siRNA transfected MCF-7 cells were subjected to colony formation assays and quantified. Data in the bar graphs represent the means ± SD values of the clone numbers for three independent experiments. **: P < 0.01. **(H)** CCK8 assays were used to assess cell proliferation for MCF-7 cells transfected with control or KPNA3 siRNA. Data in the bar graphs represent the means ± SD values of the OD450 absorption for three independent experiments. ****: P < 0.0001. **(I)** A model of KPNA3-mediated nuclear import of NPAT as essential for HLB formation and RD-histone gene transcription. Source data are available for this figure: SourceData F2.

swapping the ARM 3–5 of KPNA5 with the ARM 3–5 of KPNA3 (Fig. 5 B). Indeed, the KPNA5–3-hybrid mutant, which contains the KPNA3 ARM 3–5 region, can bind NPAT (Fig. 5 C) and block the formation of HLB (Fig. 5 D). Second, we prepared an NPAT–KPNA3 fusion mutant to mimic the permanent interaction between NPAT and KPNA3 (Fig. 5, E and F). Immunofluorescence study showed that this mutant displayed diffusive nuclear localization but no condensate formation (Fig. 5 G). These findings further solidify the conclusion that binding with KPNA3 suppresses NPAT condensation.

### The self-association sequence proximal to NLS drives NPAT condensation

To delineate the molecular mechanism underlying how KPNA3 regulates NPAT condensation, we first aimed to identify the regions that mediate the self-interaction of NAPT. KPNA3 interacts with the NLS-containing 1,031–1,427 C region of NPAT (Fig. 1, E and F). Interestingly, the C-region also binds the 431–1,030 M region of NPAT (Fig. 6 A), indicating that the interaction between M and C regions mediates the self-association of NPAT. We further tested this hypothesis by dissecting the exact interaction region inside the NPAT-C region for the binding of the NPAT-M region. Among the three truncation mutants of the C region (Fig. 6 B), only the 1,161–1,290 sequence (proximal to the NLS region of NPAT), bound the M region (Fig. 6 C). This data indicated that the 1,161–1,290 sequence acts as a C-terminal self-interaction fragment (hereafter designated C-SIF). Deletion of the C-SIF region (NPAT-C region ΔC-SIF mutant) consistently disrupted the interaction between the C region and the M region (Fig. 6 D). We next prepared an NPAT-ΔC-SIF mutant to verify if the C-SIF region is indeed important to mediate the self-association (Fig. 6 E). Co-IP analysis validated that removing the C-SIF had largely attenuated the self-association ability of NPAT (Fig. 6 F). Notably, the in vitro phase separation assay displayed impaired condensation of NPAT-ΔC-SIF (Fig. 6 G). We further created a 293T cell line stably expressing GFP-NPAT-ΔC-SIF mutant (293T-GFP-NPAT-ΔC-SIF cell). By adjusting the induction time and dosage of titration, we managed to express wild-type NPAT and NPAT-ΔC-SIF mutants in the two stable cell lines at similar levels (Fig. 6 H). Compared with wild-type NPAT, the NPAT-ΔC-SIF mutant

formed fewer foci (Fig. 6 I), and in contrast with the NPAT-ΔNLS mutant, NPAT-ΔC-SIF retained nuclear localization (Fig. 6, I and J). Together, these results revealed a self-association motif important for mediating NPAT condensation (Fig. 6 K).

### NLS-mediated KPNA3 binding sterically interferes with the self-association and condensation of NPAT

Given the proximal localization of C-SIF to the NLS, we proposed a "steric hindrance" model where the binding of KPNA3 to the NPAT-NLS would interfere with the self-association of NPAT, leading to the decondensation of NPAT. This hypothesis was supported in that the coexpression of KPNA3 largely reduced the association of the C region to the M region (Fig. 7 A). Importantly, the C region lacking the NLS displayed a stronger interaction ability with the M region (Fig. 7 B), further validating our proposed model. Notably, we found that the ARM 3–5 region of KPNA3, which mediates the association of KPNA3 to NPAT, failed to suppress NPAT condensation (Fig. 7 C). These data suggest the requirement of a certain molecular size for the NLS-anchoring molecule to block the NPAT self-association through steric effects.

To further validate the steric hindrance model, we first enlarged the size of the ARM 3–5 by adding a GFP tag or a Myc tag at this N-terminus (Fig. 7 D). This is because the Myc tag is very small (10 amino acids) while the GFP tag is much larger (27 kDa). The GFP-ARM 3–5 fusion protein, but not the Myc-ARM 3–5, did suppress NPAT condensation (Fig. 7 E) and consistently impaired the interaction between NPAT-C and M region (Fig. 7 F). These data verified the requirement of a certain molecular size for sterically blocking the NPAT condensation. In line with this finding, either KPNA3 ARM 1–5 or KPNA3 ARM 3–10 (Fig. 7 G), both of which have larger molecule sizes than ARM 3–5, could also impair the phase separation and self-association of NPAT (Fig. 7, H and I). These findings indicate that the C-SIF-dependent self-association of NPAT and HLB formation was compromised by NLS-mediated KPNA3 binding through a steric hindrance model (Fig. 8).

## Discussion

For such a protein, the function of which requires nuclear localization, the mechanism underlying the nuclear import of NPAT has remained elusive. Here, we identified that KPNA3 acts

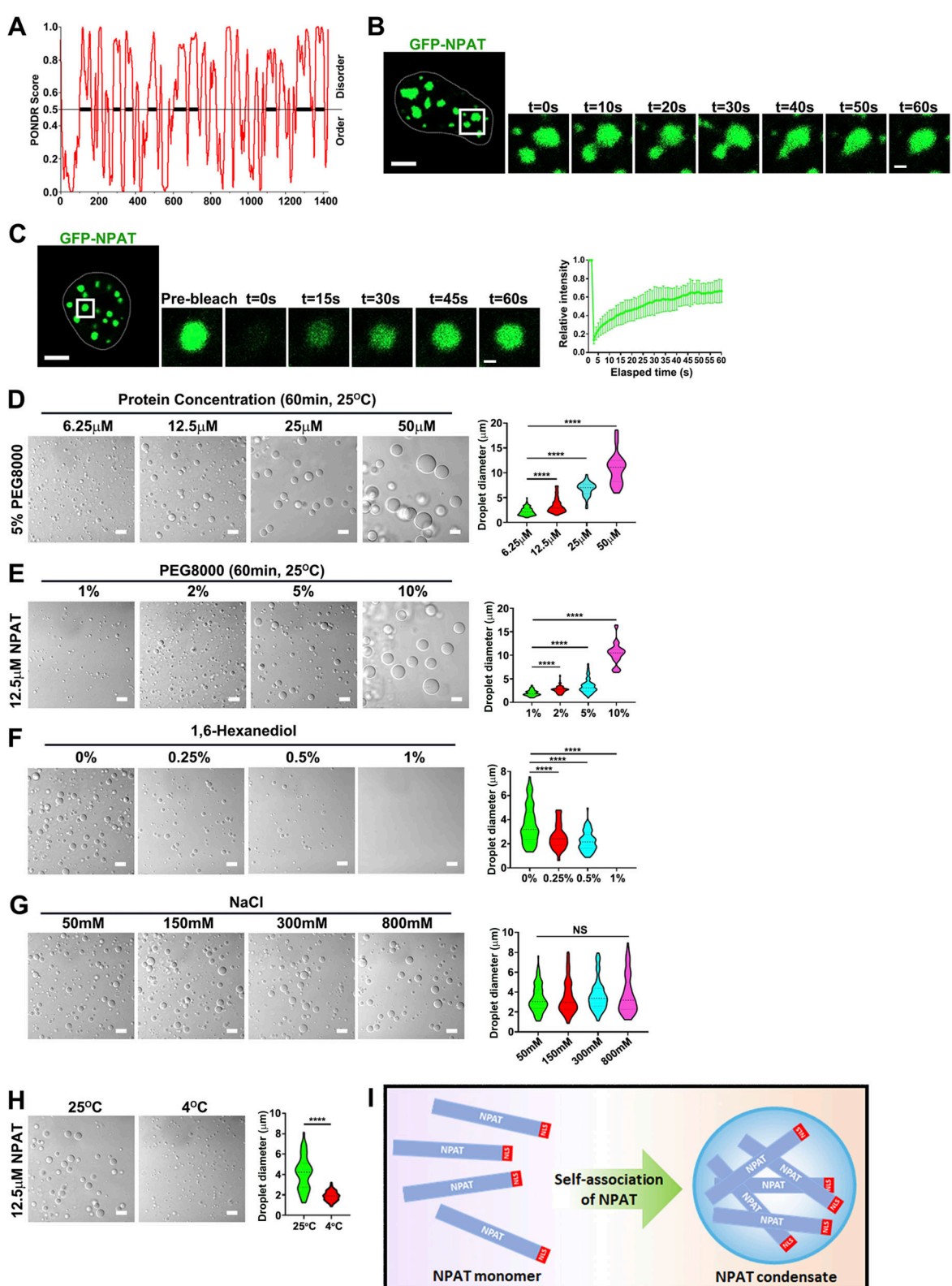

Figure 3. **NPAT undergoes phase separation. (A)** Protein sequence and disorder prediction (PONDR) of NPAT. **(B)** Time-lapse imaging showing fusion of two GFP-NPAT droplets. Scale bar, 10 µm; inset scale bar: 2 µm. **(C)** Confocal imaging and quantification of GFP-NPAT fluorescence recovery after photobleaching. Scale bar, 10 µm. Inset scale bar: 2 µm. Right panel: quantification of fluorescence intensity recovery of photobleached GFP-NPAT foci (n = 8). Data are shown as means ± SD. **(D)** DIC images of FLAG-NPAT liquid–liquid phase separation/condensation at a series of protein concentrations (6.25–50 µM) (50 mM Tris-HCl pH 7.5, 10% glycerol, 1 mM DTT, 150 mM NaCl). Scale bar, 10 µm. The size of droplets was quantified (6.25 µM: n = 104; 12.5 µM: n = 90; 25 µM: n = 48; 50 µM: n = 36). ****: P < 0.0001. **(E)** DIC images of FLAG-NPAT condensation in the presence of different concentrations of molecular crowding (1–10% wt/vol PEG-8000). Scale bar, 10 µm. The size of droplets was quantified (1%: n = 81; 2%: n = 103; 5%: n = 100; 10%: n = 51). ****: P < 0.0001. **(F)** DIC images of FLAG-NPAT

(12.5 μM) condensation with the addition of 1,6-hexanediol at indicated concentrations. Scale bar, 10 μm. The size of droplets was quantified (0%: n = 104; 0.25%: n = 80; 0.5%: n = 77; 1%: N/A). ****: P < 0.0001. **(G)** DIC images of FLAG-NPAT (12.5 μM) condensation under different salinity, as indicated. Scale bar, 10 μm. (50 mM: n = 112; 150 mM: n = 98; 300 mM: n = 88; 800 mM: n = 73). ****: P < 0.0001. **(H)** NPAT (12.5 μM) condensation at 25°C or 4°C. Scale bar, 10 μm. (25°C: n = 157; 4°C: n = 83). ****: P < 0.0001. **(I)** A model of NPAT condensation.

as a specific nuclear import receptor of NPAT. Inhibition or depletion of KPNA3 led to reduced NPAT levels in the nuclei, which subsequently impaired the NPAT condensation and HLB formation. Moreover, we showed that KPNA3 suppresses inappropriate NPAT condensation by impairing the self-association of NPAT. These findings led us to conclude that KPNA3 acts not only to mediate the nuclear import of NPAT but also to prevent the aberrant condensation of NPAT in the cytoplasm. Thus, it is of interest to examine if KPNA protein-dependent decondensation of nuclear proteins could act as a general mechanism for regulating the formation of nonmembrane organelles in nuclei. Interestingly, the ectopic expression of NPAT led to puncta formation of KPNA3 (Fig. 4, C and E), indicating that NPAT might also regulate the localization of KPNA3. This phenotype awaits further investigation in the future.

### A novel C-terminal self-association fragment mediates NPAT phase separation

Scaffold proteins act as molecular hubs for dynamically organizing signalosomes via the docking of multiple proteins to facilitate signaling transduction (DiRusso et al., 2022). Several signaling scaffolds have recently been shown to drive LLPS/condensation through conformational plasticity (DiRusso et al., 2022). In *Drosophila*, it has been shown that the formation of HLBs is mediated by the phase separation of the NPAT ortholog Mxc (Hur et al., 2020). Here, we show that NPAT also undergoes condensation in mammalian cells. Previous studies have put forward the direct interaction of both the LisH and a self-interaction facilitator (SIF) domains (both of which localize in the N-terminus of *Drosophila* Mxc) as candidates for mediating the self-association of Mxc (Terzo et al., 2015). Our data now disclose a novel C-terminal SIF domain that binds the middle region of NPAT, showing that this C-SIF domain is critical for driving NPAT condensation and HLB formation. Indeed, the C-terminus of NPAT does not form foci (Fig. 1 H), supporting the model that the C-SIF domain needs to interact with the M-region for NPAT condensation. Our findings are consistent with the previous report showing that the region of Mxc between amino acids 721 and 1,481 is also necessary for HLB assembly (Terzo et al., 2015). Thus, we concluded that it is not only the N-terminal LisH-SIF interaction but also the association between the C-SIF and middle region of NPAT that contributes to the phase separation of NPAT and HLB formation. It would therefore be of interest to further explore the regulatory machinery controlling the self-association mediated by the N-terminal LisH and SIF domains.

### KPNA3 elicits a unique steric hindrance strategy for preventing cargo condensation

Accumulating evidence has led to the conclusion that, in addition to controlling the nuclear-cytoplasmic distribution of RNA-binding proteins, nuclear import receptors also suppress protein phase separation by weakening their self-association (Guo et al., 2018; Hofweber et al., 2018; Springhower et al., 2020; Yoshizawa et al., 2018). For example, Kapβ2 not only binds the PY-NLS of FUS but also interacts with the RGG and LC regions, both of which contribute to the self-association and phase separation of FUS (Yoshizawa et al., 2018). Our study also showed that KPNA3 does not merely control the nuclear import of NPAT. Rather, it also functions in preventing aberrant condensation of NPAT in the cytoplasm. Importantly, in contrast to Kapβ2, which binds a number of regions distributed across FUS (Springhower et al., 2020), KPNA3 elicits a different strategy by specifically binding the NLS of NPAT, which is proximal to the C-SIF region that determines NPAT self-association. We believe that the KPNA3 anchors to the NPAT-NLS to sterically block the interaction between neighboring C-SIF and the M-region of NPAT. Thus, our finding revealed a unique steric hindrance model for importin-mediated decondensation. However, there remains the possibility that other cargoes of KPNA3 might indirectly contribute to the reduction of NPAT foci number. Interestingly, a recent study has also linked the cotranslational binding of importins to a wide range of pro-aggregation nascent cargoes, including ribosomal proteins and RNA-binding proteins (Seidel et al., 2023). Moreover, the condensation of TFEB was enhanced by the mutation of two positively charged amino acids in its NLS (Wang et al., 2022). These findings link with the notion that NLS acts as an anti-aggregation signal to ensure nuclear cargo remains disaggregated in the cytoplasm (Guo et al., 2018).

### Implication of the nuclear transport receptors and their cargoes in cancers

Karyopherins are essentially involved in many cellular events including signal transduction, environmental responses, and cell cycle maintenance (Li et al., 2018; Paci et al., 2021; Wing et al., 2022). Therefore impairment of Karyopherin functions leads to a series of diseases (Hazawa et al., 2020). Indeed, KPNA3-dependent nuclear localization of ataxin-3 acts as a key event in the pathogenesis of spinocerebellar ataxia type 3 (Sowa et al., 2018), while ablation of KPNA3 in mice results in abnormal sperm morphology and influences male fertility (Miyamoto et al., 2020). Notably, it has been recently revealed that dominant KPNA3 mutations cause infantile-onset hereditary spastic paraplegia in humans (Schob et al., 2021). In addition to other cargoes such as MeCP2 and NF-κB (Baker et al., 2015; Fagerlund et al., 2005; Lyst et al., 2018), NPAT is a novel cargo of KPNA3 and is involved in breast cancer and Hodgkin lymphoma (Kalla et al., 2007; Milne et al., 2017; Saarinen et al., 2011; Wright et al., 2017). These reports indicate both nuclear transport receptors and their cargoes as potential therapeutic targets. It has also been recently revealed that cofactor-derived peptides can be used to inhibit condensate-associated transcription in disease (Hirose et al., 2023). In the same way, interventions of NPAT

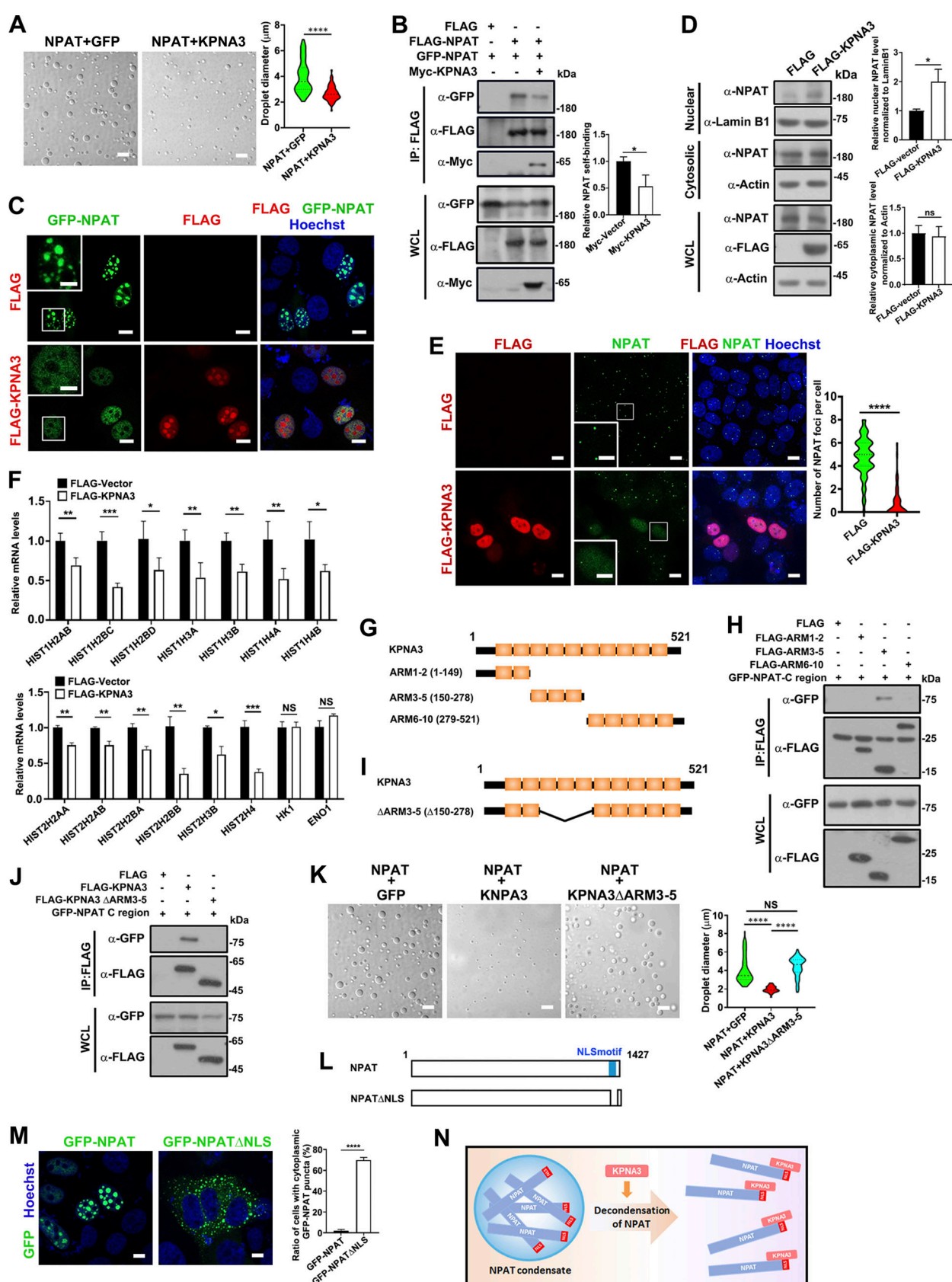

Figure 4. **KPNA3 suppresses NPAT condensation. (A)** DIC images of FLAG-NPAT (12.5 µM) condensation in the presence of GFP (12.5 µM) or KPNA3 (12.5 µM). Scale bar, 10 µm. The size of droplets was quantified (NPAT+GFP: n = 68; NPAT+KPNA3: n = 60). ****: P < 0.0001. **(B)** Coimmunoprecipitation of FLAG-tagged NPAT and GFP-NPAT in the presence or absence of Myc-KPNA3. Data in the bar graphs represent the means ± SEM values of the association of FLAG-

NPAT and GFP-NPAT. *: P < 0.05. **(C)** Confocal images of MCF-7 cells co-transfected with GFP-NPAT and FLAG vector or FLAG-KPNA3. Bar: 10 µm. The insets show the magnification of the boxed region (scale bar, 5 µm). **(D)** Cytoplasmic and nuclear fractions were prepared from MCF-7 cells transfected with FLAG vector or FLAG-KPNA3 and then subjected to western blot analysis for examination of NPAT levels. Data in the bar graphs represent the means ± SEM values of the ratio of densities for three independent experiments. *: P < 0.05, ns: not significant. **(E)** Immunofluorescence staining of endogenous NPAT in MCF-7 cells transfected with FLAG vector or FLAG-KPNA3. Scale bar, 10 µm. The insets show the magnification of the boxed region (scale bar, 5 µm). The numbers of endogenous NPAT foci per cell were quantified (FLAG vector: n = 111; FLAG-KPNA3: n = 87). ****: P < 0.0001. **(F)** MCF-7 cells transfected with a control vector or FLAG-KPNA3 were subject to qPCR analysis for transcription levels of the indicated RD histone genes and two control genes. Data in the bar graphs represent the means ± SEM values of the relative transcription levels for three independent experiments. *: P < 0.05, **: P < 0.01, ***: P < 0.001, ****: P < 0.0001, ns: not significant. **(G)** Schematic diagram of KPNA3 and its truncation mutants. **(H)** Coimmunoprecipitation of FLAG-tagged KPNA3 truncation mutants and GFP-NPAT C-region. **(I)** Schematic diagram of KPNA3 and its ARM 3–5 deletion mutant (ΔARM3–5). **(J)** Coimmunoprecipitation of FLAG-tagged KPNA3 wild type or the ΔARM3–5 mutant and GFP-NPAT C-region. **(K)** DIC images of FLAG-NPAT (12.5 µM) condensation in the presence of GFP, KPNA3, or KPNA3 ΔARM3–5 mutant at the concentration of 12.5 µM. Scale bar, 10 µm. The size of droplets was quantified (NPAT+GFP: n = 85; NPAT+KPNA3: n = 44; NPAT+KPNA3 ΔARM3–5: n = 70). ****: P < 0.0001. **(L)** Schematic diagram of NPAT and its NLS deletion mutant (NPAT-ΔNLS). **(M)** Confocal images of MCF-7 cells transfected with GFP-NPAT or GFP-NPAT-ΔNLS. Bar: 5 µm. The ratio of cells with cytoplasmic NPAT foci was quantified for three independent experiments. ****: P < 0.0001. **(N)** A model of KPNA3-mediated decondensation of NPAT. Source data are available for this figure: SourceData F4.

## Materials and methods

### Culture and maintenance of cells
MCF-7 cell line (RRID: CVCL_0031) was obtained from the Chinese Academy of Sciences Cell Bank. The HEK293T cell line (RRID: CVCL_0063) was from the American Type Culture Collection. MCF-7 cells were grown in DMEM (high glucose) supplemented with 15% (vol/vol) or 10% fetal bovine serum respectively. 293T cells were grown in RPMI-1640 medium supplemented with 10% (vol/vol) fetal bovine serum, 2 mM L-glutamine, 100 U/ml penicillin, and 100 mg/ml streptomycin (all from Hyclone Laboratories). Cells in 6-well plates were transfected with lipofectamine 2000 (11668019; Invitrogen) according to the manufacturer's protocol.

### Construction of expression plasmids
The coding sequence of KPNA family proteins, NPAT, and truncation mutants was inserted into a pXJ40 expression vector with GFP, FLAG, Myc, or HA tag. NPAT C-SIF deletion, NLS deletion, and KPNA3 ARM3–5 deletion mutants were constructed using 2×MultiF Seamless Assembly Mix (RK21020; Abclonal) according to the manufacturer's protocol. All the DNA sequences were amplified with 2×Phanta Max Master Mix (Dye Plus) (P525-01; Vazyme) according to the manufacturer's protocol. The sequences of the primers for gene cloning are listed in Table S1.

### Generation of an inducible stable cell line
A tet-on system was used for 293T or MCF-7 cells to generate inducible stable cell lines as previously described (Gao et al., 2022). Cells were cotransfected with an HP216 vector and HP138-GFP-NPAT-related vectors. Cell lines were obtained through 5 µg/ml puromycin selection. Cells were treated with 500 ng/ml doxycycline (60204ES03; YEASEN) for 1 day to induce the expression of proteins. Western blot was then performed to validate expression levels.

### Estimation of endogenous NPAT protein concentrations
The concentration of endogenous NPAT protein was measured following the previously reported protocol (Gao et al., 2022).

Briefly, quantification was based on western blot densitometry analysis performed on cell lysates and purified FLAG-NPAT protein. MCF-7 cells or HEK293T cells were lysed in WB lysis buffer with protease inhibitors and subjected to western blot with purified FLAG-NPAT protein. After densitometry analysis of western blot results using Fiji (RRID: SCR_002285), we plotted the band density against purified FLAG-NPAT concentrations.

### Antibodies
The following antibodies were used for western blotting and immunofluorescence: mouse anti-GFP (T0005, RRID: AB_2839413; Affinity Biosciences), mouse anti-FLAG (30503ES60, RRID: AB_3095723; Yeasen Biotechnology), rabbit anti-HA (0906-1, RRID:AB_3068712; HuaAn Biotechnology), rabbit anti-FLAG (F7425, RRID: AB_439687; Sigma-Aldrich), rabbit anti-Lamin B1 (R1508-1, RRID:AB_3073316; HuaAn Biotechnology), rabbit anti-Myc (R1208-1, RRID:AB_3073204; HuaAn Biotechnology), goat KPNA3 (ab6038, RRID: AB_305247; Abcam), mouse anti-NPAT (611344, RRID: AB_398866; BD Biosciences), rabbit anti-NPAT (A302-772A, RRID: AB_10630262; Bethyl), mouse anti-FLASH (ab8402, RRID: AB_306556; Abcam), and mouse anti-Actin (M1210-2, RRID: AB_3073045; HuaAn Biotechnology). The HRP-conjugated secondary antibodies used were goat anti-mouse second antibody (115-035-003; Jackson) and goat anti-rabbit second antibody (111-035-003; Jackson). The fluorescent secondary antibodies were follows: goat anti-mouse-Alexa Fluor 546 (A-11030, RRID: AB_2737024; Thermo Fisher Scientific), goat anti-mouse-Alexa Fluor 488 (A-11001, RRID: AB_2534069; Thermo Fisher Scientific), goat anti-rabbit-Alexa Fluor 488 (A-11034, RRID: AB_2576217; Thermo Fisher Scientific), goat anti-rabbit-Alexa Fluor 546 (A-11035, RRID: AB_2534093; Thermo Fisher Scientific), and goat anti-rabbit-Alexa Fluor 633 (A-21071, RRID: AB_141419; Thermo Fisher Scientific).

### Colony formation assay
$1 \times 10^3$ MCF-7 cells were seeded in triplicate in 6-cm dishes and maintained in DMEM medium supplemented with 10% fetal bovine serum. The growth medium was changed every 3 days. After 7 days, the resulting colonies were rinsed with PBS, fixed with 4% paraformaldehyde for 10 min, and stained with Crystal Violet Staining Solution (E607309-100; Sangon Biotech).

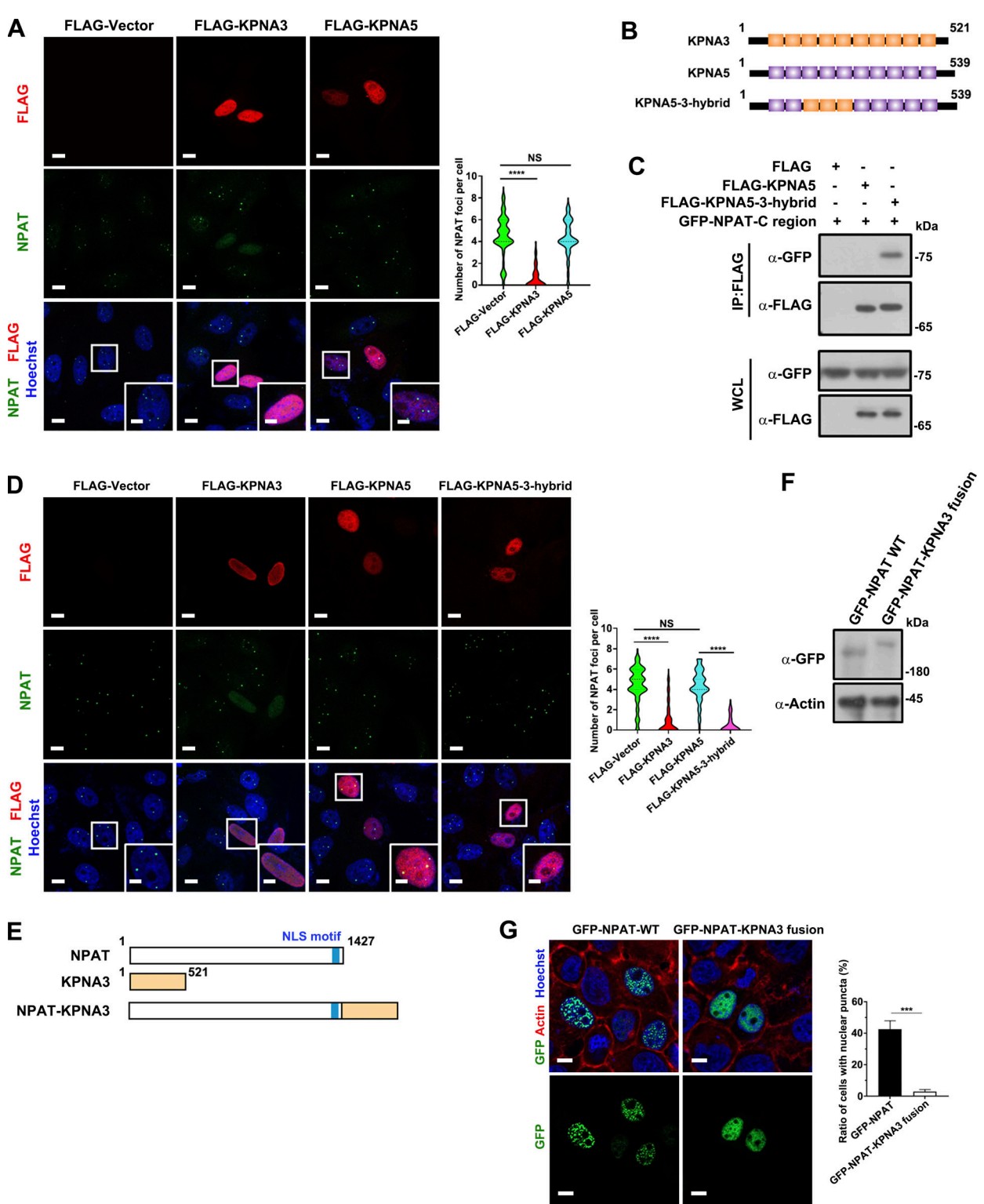

Figure 5.    **The binding to KPNA3 suppresses the condensation of NPAT. (A)** Immunofluorescence staining of endogenous NPAT in MCF-7 cells transfected with FLAG vector, FLAG-KPNA3, or FLAG-KPNA5. Scale bar, 10 μm. The insets show the magnification of the boxed region (scale bar, 5 μm). The numbers of endogenous NPAT foci per cell were quantified (FLAG vector: $n$ = 109; FLAG-KPNA3: $n$ = 102; FLAG-KPNA5: $n$ = 107). ****: P < 0.0001, ns: not significant. **(B)** Schematic diagram of KPNA3, KPNA5, and KPNA5-3-hybrid mutant. **(C)** Coimmunoprecipitation of GFP-NPAT C-region and FLAG-tagged KPNA3 or KPNA5-3-hybrid mutant. **(D)** Immunofluorescence staining of endogenous NPAT in MCF-7 cells transfected with FLAG vector, FLAG-KPNA3, FLAG-KPNA5, or FLAG-KPNA5-3-hybrid. Scale bar, 10 μm. The insets show the magnification of the boxed region (scale bar, 5 μm). The numbers of endogenous NPAT foci per cell were quantified (FLAG vector: $n$ = 104; FLAG-KPNA3: $n$ = 92; FLAG-KPNA5: $n$ = 95; FLAG-KPNA5-3-hybrid: $n$ = 87). ****: P < 0.0001, ns: not significant. **(E)** Schematic diagram of NPAT, KPNA3, and NPAT-KPNA3 fusion mutant. **(F)** Western blot analysis of GFP-NPAT and GFP-NPAT-KPNA3 fusion mutant expression in MCF-7 cells. **(G)** Confocal images of MCF-7 cells transfected with GFP-NPAT or GFP-NPAT-KPNA3 fusion. Bar: 10 μm. The ratio of cells with nuclear NPAT foci was quantified for three independent experiments. ***: P < 0.001. Source data are available for this figure: SourceData F5.

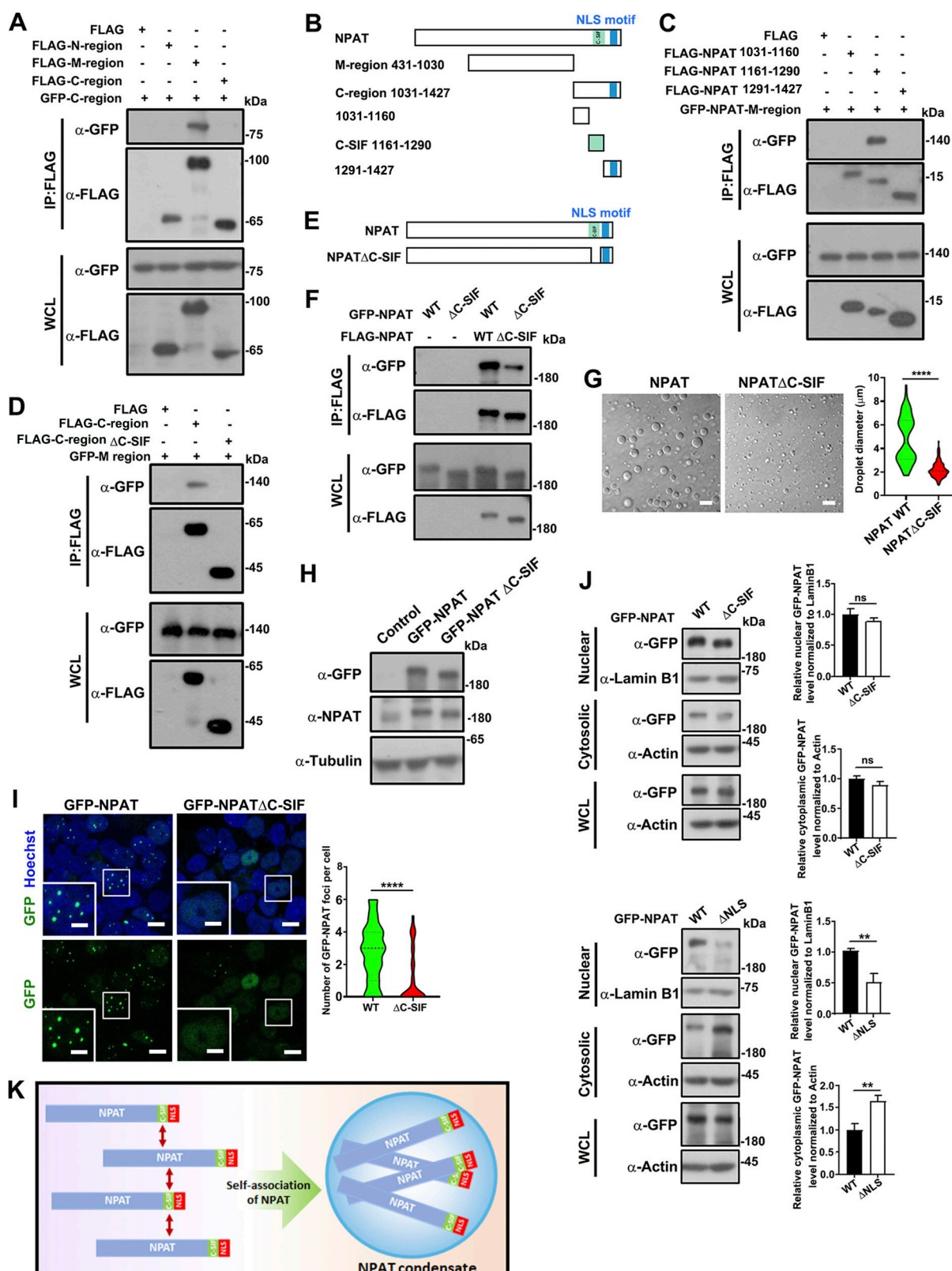

Figure 6. **NPAT condensation is dependent on a self-association sequence proximal to NLS. (A)** Coimmunoprecipitation of FLAG-tagged NPAT truncation mutants and GFP-NPAT C-region. **(B)** Schematic diagram of NPAT and its truncation mutants. **(C)** Coimmunoprecipitation of FLAG-tagged NPAT truncation mutants as shown in B and GFP-NPAT M-region. **(D)** Coimmunoprecipitation of FLAG-tagged NPAT C-region wild type or the C-region ΔC-SIF mutant and GFP-NPAT M-region. **(E)** Schematic diagram of NPAT and its ΔC-SIF deletion mutant. **(F)** Coimmunoprecipitation of FLAG-NPAT and GFP-NPAT or FLAG-NPAT ΔC-SIF and GFP-NPAT ΔC-SIF. **(G)** DIC images of FLAG-NPAT (12.5 μM) or FLAG-NPAT ΔC-SIF (12.5 μM) condensation. Scale bar, 10 μm. The size of droplets was quantified (FLAG-NPAT: n = 70; FLAG-NPAT ΔC-SIF: n = 105). ****: P < 0.0001. **(H)** Expression levels of GFP-NPAT and GFP-NPAT-ΔC-SIF mutant in the 293T-

GFP-NPAT and 293T-GFP-NPAT-ΔC-SIF stable cell lines. **(I)** Confocal images of GFP foci in the 293T-GFP-NPAT and 293T-GFP-NPAT-ΔC-SIF stable cell lines. Bar: 10 µm. The insets show the magnification of the boxed region (scale bar, 5 µm). The numbers of foci per cell were quantified (wild-type NPAT: $n = 76$; NPAT-ΔC-SIF: $n = 78$). ****: $P < 0.0001$. **(J)** Cytoplasmic and nuclear fractions were prepared from the 293T-GFP-NPAT and 293T-GFP-NPAT-ΔC-SIF or 293T-GFP-NPAT-ΔNLS stable cell lines and then subjected to western blot analysis for examination of GFP-tagged NPAT levels. Data in the bar graphs represent the means ± SEM values of the ratio of densities for three independent experiments. **: $P < 0.01$, ns: not significant. **(K)** A model of C-SIF-mediated self-association driving NPAT condensation. Source data are available for this figure: SourceData F6.

### Cell proliferation assay

$1 \times 10^3$ MCF-7 cells were seeded in 96-well plates in triplicate and treated with KPNA3 siRNA for 72 h. Cell growth was measured using a Cell Counting Kit-8 (CCK-8) (HY-K0301; MCE) every 24 h. Briefly, MCF-7 cells were washed twice with PBS and mixed with a reaction solution containing CCK8. After incubation at 37°C for 1.5 h, the absorption values of OD450 were read using a microplate reader (SpectraMax iD5; Molecular Devices).

### Quantitative RT-qPCR

As previously described, total RNA was isolated using an AFT-Spin Tissue/Cell Fast RNA Extraction Kit (RK30120; Abclonal) (He et al., 2019). Reverse transcription was performed using an ABScript III RT Master Mix for qPCR with gDNA Remover (RK20429; Abclonal). The qPCR reactions were carried out using 2×Universal SYBR Green Fast qPCR Mix (RM21203; Abclonal) on Roche LightCycler 480 Real-Time PCR System (RRID: SCR_018626). Each RT-qPCR experiment was technically replicated and performed with four biological replicates. All respective primers are listed in Table S2.

### RNA interference

MCF-7 cells at 40–50% confluency were transfected with 100 nM of siRNA using Lipofectamine RNAiMAX (13778150; Invitrogen) as previously described (Cong et al., 2020). The sequence of siRNAs is:

KPNA3 siRNA-1: 5′-GGAAACAATGCGAAGACAT-3′; KPNA3 siRNA-2: 5′-CCACCGATTGATGACTTAA-3′; NPAT siRNA: 5′-GGGTTTGCGAAGTGAGAAA-3′.

### Protein expression and purification

The FLAG-tagged vectors were transfected into 293T cells. 2 days later, the cells were harvested by centrifugation at 14,000 rpm for 10 min at 4°C and lysed with HEPES buffer (150 mM sodium chloride, 50 mM HEPES, pH 7.4, 5 mM EDTA, 1% [wt/vol] sodium deoxycholate, 1% [vol/vol] Triton X-100, 0.2% sodium fluoride, 0.1% sodium orthovanadate, and protease inhibitor cocktail [B14001; Selleck Chemicals]). Anti-FLAG M2 gel beads (B23102; Bimake) were then added and incubated on a rotary shaker at 4°C for 2 h. M2 gel beads were harvested by centrifugation at 3,000 rpm for 1 min and washed in the HEPES buffer. The FLAG-tagged protein was purified by competition using 3×FLAG peptide (HY-P0319; MCE). Briefly, M2 beads were resuspended with 1.5 mg/ml 3×FLAG peptide buffer and incubated at 4°C for 2 h. After centrifugation at 5,000 rpm for 1 min, the supernatant was concentrated to 100 mg/ml by using a Microcon-100-kDa Centrifugal Filter Unit with Ultra-100 membrane (UFC810024; Millipore) with storage buffer (50 mM Tris-HCl pH 7.5, 37 mM NaCl, 1 mM EDTA, 5 mM DTT) and then stored at −80°C.

### In vitro phase separation assays

The purified proteins were added to the phase separation Buffer (50 mM Tris-HCl pH 7.5, 10% glycerol, 1 mM DTT, and 10% PEG-8000). For measuring the effect of GFP, KPNA, or KPNA mutant on NPAT condensation, different proteins were mixed at the start point of the assay. The concentration of NaCl was adjusted to the indicated concentrations. The protein solution was loaded onto an 8-well chamber (C8-1.5H-N; Cellvis) for 5 min at room temperature and then imaged using a Zeiss LSM 800 confocal microscope with a 63× objective (Carl Zeiss). For temperature-mediated phase separation, the 8-well chamber was firstly kept at 25°C for 5 min and then shifted to 4°C for 5 min. For 1,6-hexanediol (H0099; TCI)-mediated phase separation regulation, 1,6-hexanediol was present at the indicated concentrations.

### Fluorescence recovery after photobleaching (FRAP)

FRAP experiments were performed on a Zeiss LSM 800 microscope with a 63× oil immersion objective. MCF-7 cells were seeded onto 8-well chamber slides (C8-1.5H-N; Cellvis). Cells were transfected with GFP-NPAT plasmids. After a 24-h incubation, GFP-NPAT droplets were photobleached using a laser intensity of 80% at 488 nm (for GFP) and recovery was recorded for the indicated time. The prebleached fluorescence intensity was normalized to one and the signal after bleaching was normalized to the prebleached level.

### Immunoprecipitation studies and western blot analyses

Control cells or cells transfected with expression plasmids were lysed in lysis buffer (150 mM sodium chloride, 0.25 mM EDTA, 50 mM Tris, pH 7.3, 1% [wt/vol] sodium deoxycholate, 0.2% sodium fluoride, 1% [vol/vol] Triton X-100, 0.1% sodium orthovanadate, and protease inhibitor cocktail [B14001; Selleck Chemicals]). Lysates were immunoprecipitated (IP) with anti-FLAG M2 beads (B23102; Bimake) or protein A/G agarose resin (36403ES25; YEASEN). Denatured protein samples were separated by SDS-PAGE gels and then transferred to a FluoroTrans polyvinylidene difluoride membrane (#BSP0106; Pall). After blocking with 5% (wt/vol) bovine serum albumin for 30 min at room temperature, the membrane was hybridized with corresponding primary and secondary antibodies.

### Immunofluorescence and direct fluorescence studies

Cells were seeded on coverslips in a 6-well plate and transfected with various expression constructs for 24–36 h. They were then stained for immunofluorescence detection using confocal fluorescence microscopy or directly visualized for cells expressing GFP-tagged proteins as previously described (Rao et al., 2021). Images were collected at room temperature using a 63 × 1.4 NA or 20× objective lens using appropriate laser excitation on an

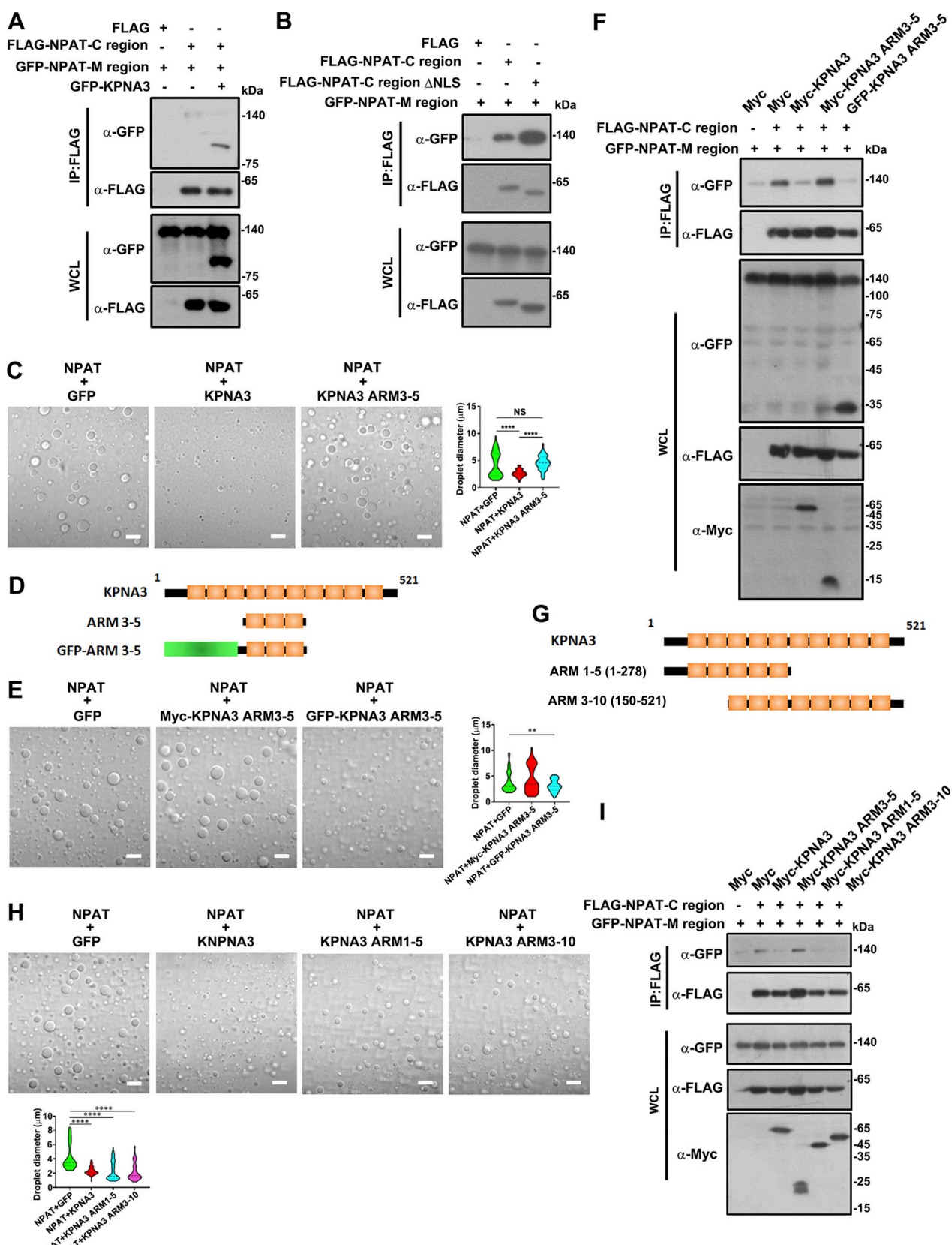

Figure 7. **Steric binding of KPNA3 interferes with the self-association and condensation of NPAT. (A)** Coimmunoprecipitation of FLAG-tagged C-region and GFP-M-region in the presence or absence of GFP-KPNA3. **(B)** Coimmunoprecipitation of FLAG-tagged C-region or its ΔNLS mutant and GFP-M-region. **(C)** DIC images of FLAG-NPAT (12.5 μM) condensation in the presence of GFP, KPNA3, or KPNA3 ARM 3–5 mutants at a concentration of 12.5 μM. Scale bar, 10 μm. The size of droplets was quantified (NPAT+GFP: $n$ = 62; NPAT+KPNA3: $n$ = 56; NPAT+KPNA3 ARM3–5: $n$ = 85). ****: P < 0.0001, ns: not significant.

**(D)** Schematic diagram of KPNA3, ARM 3–5 and GFP-ARM 3–5. **(E)** DIC images of FLAG-NPAT (12.5 µM) condensation in the presence of GFP, FLAG-KPNA3 ARM 3–5, or GFP-KPNA3 ARM 3–5 mutant at a concentration of 12.5 µM. Scale bar, 10 µm. The size of droplets was quantified (NPAT+GFP: n = 78; NPAT+myc-KPNA3 ARM3–5: n = 74; NPAT+GFP-KPNA3 ARM3–5: n = 53). **: P < 0.01. **(F)** Coimmunoprecipitation of FLAG-tagged C-region and GFP-M-region in the presence of Myc-KPNA3, Myc-AMR 3–5, or GFP-ARM 3–5. **(G)** Schematic diagram of KPNA3, ARM 1–5, and ARM 3–10. **(H)** DIC images of FLAG-NPAT (12.5 µM) condensation in the presence of GFP, FLAG-KPNA3, FLAG-ARM 1–5, or FLAG-ARM 3–10 mutant at a concentration of 12.5 µM. Scale bar, 10 µm. The size of droplets was quantified (NPAT+GFP: n = 74; NPAT+KPNA3: n = 80; NPAT+KPNA3 ARM1-5: n = 97; NPAT+KPNA3 ARM3-10: n = 110). ****: P < 0.0001. **(I)** Coimmunoprecipitation of FLAG-tagged C-region and GFP-M-region in the presence of GFP, Myc-KPNA3, Myc-ARM 1–5, or Myc-ARM 3–10 mutant. Source data are available for this figure: SourceData F7.

LSM800 Meta laser-scanning confocal microscope (RRID: SCR_015963; Carl Zeiss). The detector gain was first optimized by sampling various regions of the coverslip and then fixed for each specified channel. Once set, the detector gain value was kept constant throughout the image acquisition process. Images were analyzed using Zeiss LSM Image Examiner Software. For endogenous NPAT foci image acquisition, and Z stack images were captured with 0.12-µm interval. Images were processed using ZEN software (Black Edition) and the maximum intensity projection method.

### Three-dimensional (3D) rendering and sphericity measurement

MCF-7 cells transiently transfected with GFP-NPAT were fixed with 3% paraformaldehyde. Z stack images were acquired using a Zeiss LSM 800 confocal microscope. The step size was 0.12 µm. 3D rendering was performed using Imaris software (RRID: SCR_007370). Sphericity was calculated using Imaris software.

### Nuclear and cytoplasmic protein extraction

After Ivermectin (MK933; MCE) treatments or KPNA3 siRNA knockdown, cells were washed and harvested in cold PBS. The cytoplasmic and nuclear fractions were then separated by using a Nuclear and Cytoplasmic Protein Extraction Kit (P0028; Beyotime) according to the manufacturer's instructions. Samples were run in SDS/PAGE gels and analyzed by western blotting with the indicated antibodies.

### Statistical analysis

Statistical analyses were performed using GraphPad Prism 8.0.2 (RRID: SCR_002798; GraphPad Software, Inc.). Results are presented as mean ± SEM or mean ± SD. Statistical significance was determined as indicated in the figure legends: *: $P < 0.05$, **: $P < 0.01$, ***: $P < 0.001$, ****: $P < 0.0001$. The data distribution was first checked using a Shapiro–Wilk normality test, Kolmogorov–Smirnov test, and D'Agostino and Pearson omnibus normality test. For comparison between two normally distributed datasets, an F-test was used to determine the variances. If the variances are similar, a two-tailed unpaired Student's test was used ($P > 0.05$). A two-tailed unpaired Student's $t$ test with Welch's correction was used when variances were shown as different via the F test ($P < 0.05$). If the data did not fit a normal distribution, a Mann–Whitney test was used. If the variation among three or more groups was minimal, ANOVA followed by Dunnett's

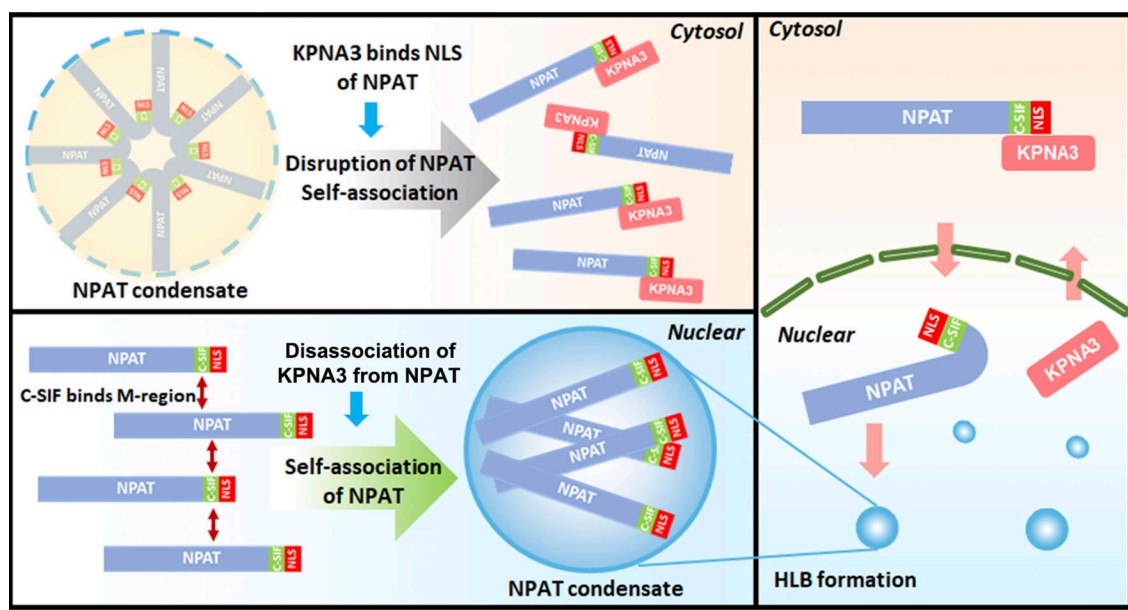

Figure 8.   **Schematic model for KPNA3-mediated NPAT decondensation and HLB remodeling.** We found that KPNA3 determines the nuclear import of NPAT by interacting with the C-terminus NLS region. A C-SIF region proximal to NLS is essential for mediating the self-association of NPAT to drive the condensation of NPAT in nuclei. In the cytoplasm, NLS-mediated KPNA3 association sterically blocks the C-SIF motif-dependent NPAT self-association, which ensures the nuclear import of NPAT and prevents aberrant cytoplasmic HLB formation.

post-test or Tukey's post hoc test was applied to compare multiple groups.

### Online supplemental material

Fig. S1 shows the establishment of a stable cell line expressing GFP-NPAT in an inducible manner. Fig. S2 shows that ivermectin treatment reduces the nuclear import of NPAT. Fig. S3 shows the sphericity of NPAT droplets and quantification of endogenous NPAT concentration. Fig. S4 shows the comparison of NPAT and KPNA3 concentration. Table S1 shows primers for gene cloning. Table S2 shows the primers for RT-qPCR.

### Data availability

The data are available from the corresponding author upon reasonable request.

## Acknowledgments

We thank Dr. Haixin Yuan from Fudan University for providing the KPNAs plasmids, Ms. Yanwei Li, Xiaonan Linzhao, Shuangshuang Liu, and Guifeng Xiao from the core facility platform, and Qin Han from Center of Cryo-Electron Microscopy of Zhejiang University School of Medicine for their technical support.

This work was supported by the National Natural Science Foundation of China (32070821, T2121004, 81871760, 32270720, 32100671, and 82072201), the National Key R&D Program of China (2021YFA1100500 and 2018YFA0800403), grants from the Science and Technology Commission of Zhejiang Province (2022R52040), and the China Postdoctoral Science Foundation (2021M702848 and 2019M662034).

Author contributions: S.B. Xu: Formal analysis, Investigation, Methodology, Project administration, Resources, Validation, Visualization, Writing - review & editing, X.K. Gao: Formal analysis, Investigation, Resources, Validation, Visualization, H.D. Liang: Investigation, Methodology, Validation, Visualization, X.X. Cong: Investigation, Methodology, Validation, Visualization, X.Q. Chen: Investigation, Methodology, Validation, Visualization, W.K. Zou: Investigation, J.L. Tao: Investigation, Methodology, Validation, Visualization, Z.Y. Pan: Investigation, Methodology, Validation, Visualization, J. Zhao: Investigation, Methodology, Validation, Visualization, M. Huang: Project administration, Visualization, Z. Bao: Conceptualization, Resources, Supervision, Validation, Writing - original draft, Writing - review & editing, Y.T. Zhou: Conceptualization, Funding acquisition, Project administration, Resources, Supervision, Validation, Visualization, Writing - original draft, Writing - review & editing, L.L. Zheng: Conceptualization, Funding acquisition, Resources, Supervision, Validation, Visualization, Writing - original draft, Writing - review & editing.

Disclosures: The authors declare that no competing interests exist.

Submitted: 7 January 2024

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

# Supplemental material

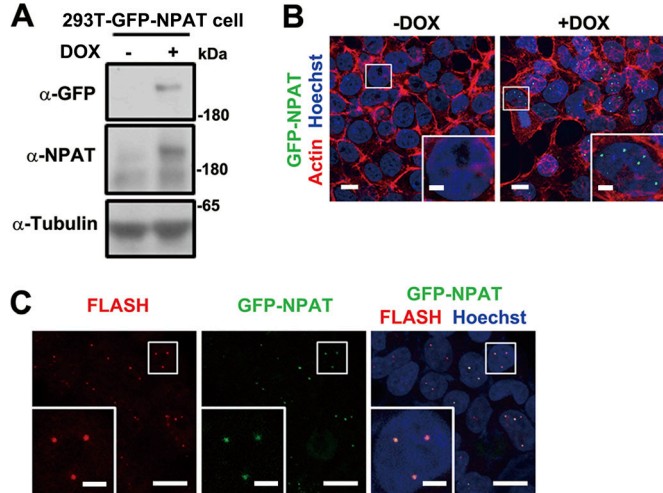

Figure S1. **Establishment of a stable cell line expressing GFP-NPAT in an inducible manner. (A)** Immunoblot analysis of NPAT expression levels in untreated or Dox-induced 293T-GFP-NPAT cells. **(B)** Confocal images of GFP-NPAT foci in untreated or Dox-induced 293T-GFP-NPAT cells. Scale bar, 10 μm. The insets show the magnification of the boxed region (scale bar, 5 μm). **(C)** Confocal images of endogenous FLASH and GFP-NPAT foci in Dox-induced 293T-GFP-NPAT cells. Scale bar, 10 μm. The insets show the magnification of the boxed region (scale bar, 2.5 μm). Source data are available for this figure: SourceData FS1.

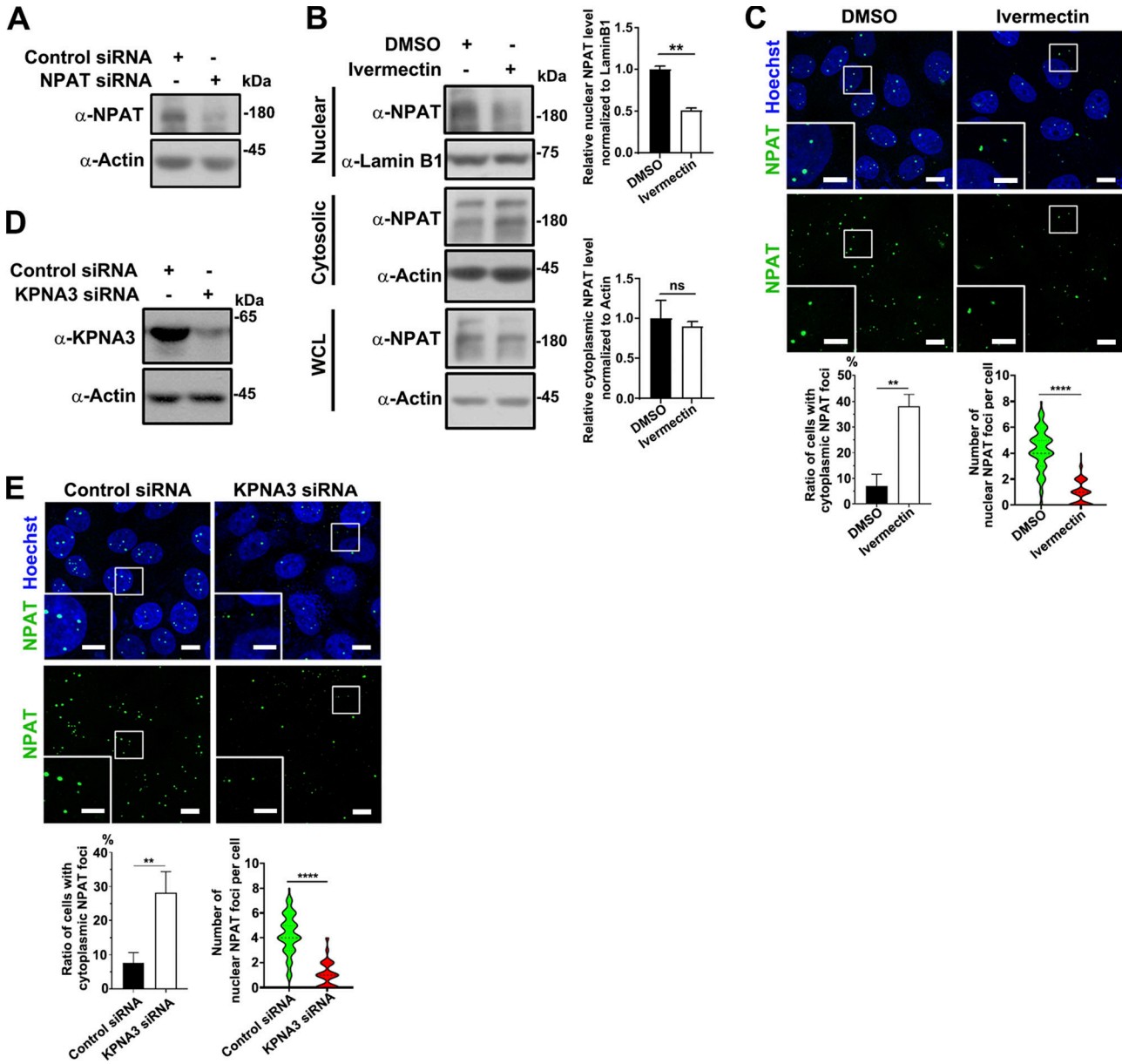

Figure S2. **Ivermectin treatment reduces the nuclear import of NPAT. (A)** Western blot analysis of endogenous NPAT expression in MCF-7 cells transfected with control or NPAT siRNA. **(B)** Cytoplasmic and nuclear fractions were prepared from control or ivermectin-treated MCF-7 cells and then subjected to western blot analysis to examine NPAT levels. Data in the bar graphs represent the means ± SEM values of the ratio of densities for three independent experiments. **: P < 0.01, ns: not significant. **(C)** Immunofluorescence staining of endogenous NPAT in MCF-7 cells treated with 50 μM ivermectin. Scale bar, 10 μm. The insets show the magnification of the boxed region (scale bar, 5 μm). The ratio of cells with NPAT foci in cytoplasm was quantified and data in the bar graphs represent the means ± SD value for three independent experiments. **: P < 0.01. The nuclear NPAT foci per cell were quantified (control: n = 170; ivermectin: n = 166). ****: P < 0.0001. **(D)** KPNA3 siRNA reduced the expression levels of endogenous KPNA3. **(E)** Immunofluorescence staining of endogenous NPAT in MCF-7 cells transfected with control or KPNA3 siRNA for 72 h. Scale bar, 10 μm. The insets show the magnification of the boxed region (scale bar, 5 μm). The ratio of cells with NPAT foci in cytoplasm was quantified and data in the bar graphs represent the means ± SD value for three independent experiments. **: P < 0.01. The nuclear NPAT foci per cell were quantified (control siRNA: n = 160; KPNA3 siRNA: n = 159). ****: P < 0.0001. Source data are available for this figure: SourceData FS2.

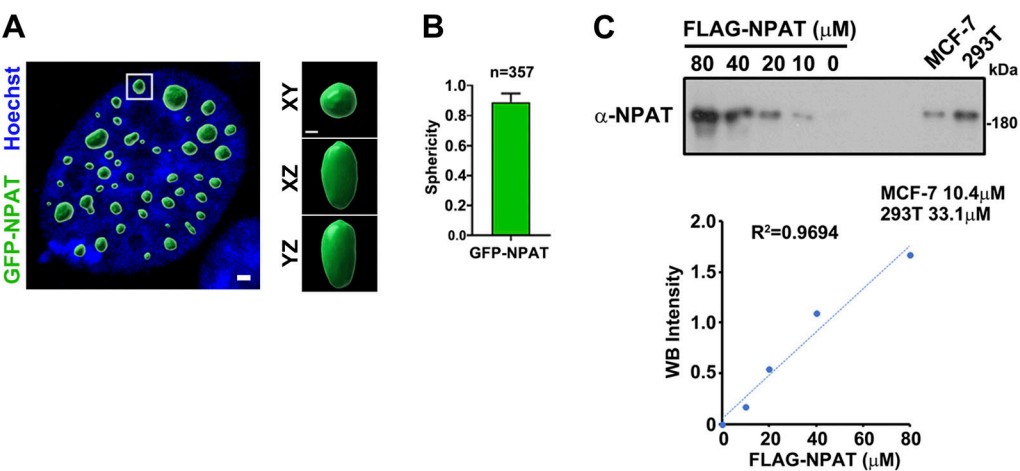

Figure S3.   **Sphericity of NPAT droplets and Quantification of endogenous NPAT concentration. (A)** Rendered 3D shapes of NPAT droplets in a cell. The panels show the XY, XZ, and YZ planes. Scale bar, 2 µm. The insets show the magnification of the boxed region (scale bar, 0.5 µm). **(B)** A plot showing the sphericity of NPAT droplets (*n* = 357). The quantification result is shown as mean ± SD. **(C)** Quantification result of endogenous NPAT protein concentrations in MCF-7 and 293T cells based on immunoblot densitometry analysis performed on cell lysates and purified FLAG-NPAT protein. Source data are available for this figure: SourceData FS3.

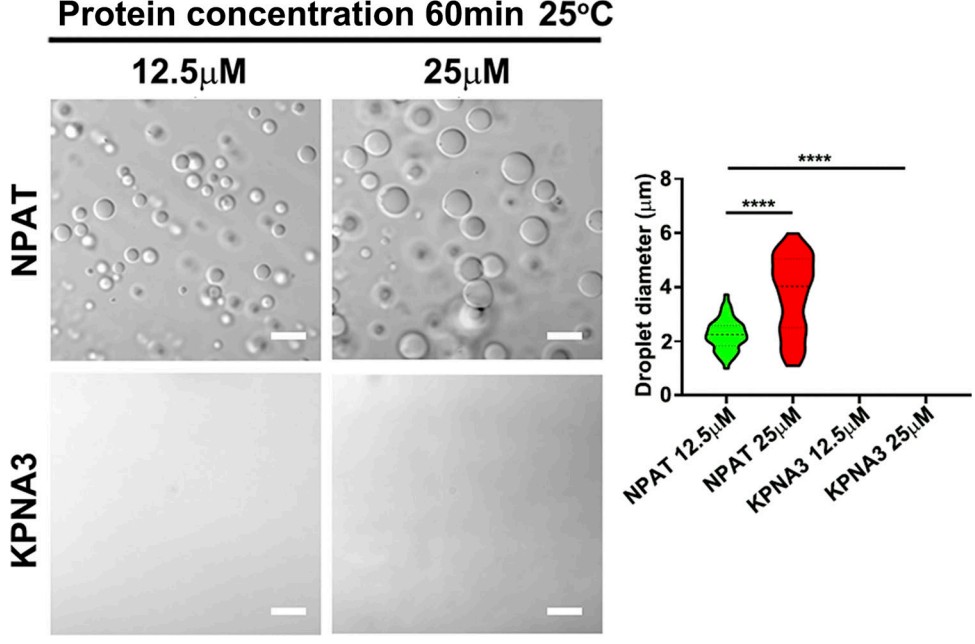

Figure S4.   **Comparison of NPAT and KPNA3 concentration.** DIC images of FLAG-NPAT and FLAG-KPNA3 condensation at a series of protein concentrations (12.5–25 µM). The size of droplets was quantified (NPAT 12.5 µM: *n* = 51; NPAT 25 µM: *n* = 38; KPNA3: N/A). ****: P < 0.0001.

**Provided online are Table S1 and Table S2. Table S1 shows primers for gene cloning. Table S2 shows primers for RT-qPCR.**

