## [Peer Review File · The Journal of Cell Biology]

KPNA3 regulates histone locus body formation by modulating condensation and nuclear import of NPAT

Shui Bo Xu, Xiu Kui Gao, Hao Di Liang, Xiao Xia Cong, Xu Qi Chen, Wen Kai Zou, Jia Li Tao, Zhao Yuan Pan, Jiao Zhao, Man Huang, Zhang Bao, Yiting Zhou, and Li Ling Zheng

Corresponding Author(s): Li Ling Zheng, Second Affiliated Hospital of Zhejiang University

Review Timeline:

Submission Date:	2024-01-07
Editorial Decision:	2024-02-16
Revision Received:	2024-07-30
Editorial Decision:	2024-08-28
Revision Received:	2024-09-06

Monitoring Editor: Michael Rout

Scientific Editor: Dan Simon

Transaction Report:

DOI: <https://doi.org/10.1083/jcb.202401036>

February 16, 2024

Re: JCB manuscript #202401036

Dr. Li Ling Zheng
Second Affiliated Hospital of Zhejiang University

Dear Dr. Zheng,

Thank you for submitting your manuscript entitled "KPNA3 regulates histone locus body by modulating the condensation and nuclear import of NPAT." The manuscript was assessed by expert reviewers, whose comments are appended to this letter. Overall, the reviewers were enthusiastic and support publication in JCB, but all require additional experimentation and manuscript edits in order to address their concerns. We invite you to submit a revision if you can address the reviewers' key concerns, as outlined here.

The main challenge is distinguishing the effects of mislocalization due to impaired import from the impact of KPNA3 binding on NPAT condensation, and as such we agree with Reviewers 1&3 that it would be important to confirm that NPAT condensation is regulated by KPNA3 in cells, as you showed occurs in vitro. The reviewers suggested several experiments to demonstrate this, including the creation of NPAT mutants and NLS fusions, and the examination of co-overexpression effects. While we do not expect you to do all of these, at least some new data is necessary to show that in cells NPAT behaves similarly to in vitro.

GENERAL GUIDELINES:

Text limits: Character count for an Article is < 40,000, not including spaces. Count includes title page, abstract, introduction, results, discussion, and acknowledgments. Count does not include materials and methods, figure legends, references, tables, or supplemental legends.

Figures: Articles may have up to 10 main text figures. Figures must be prepared according to the policies outlined in our Instructions to Authors, under Data Presentation, <https://jcb.rupress.org/site/misc/ifora.xhtml>. All figures in accepted manuscripts will be screened prior to publication.

Supplemental information: There are strict limits on the allowable amount of supplemental data. Articles may have up to 5 supplemental figures. Up to 10 supplemental videos or flash animations are allowed. A summary of all supplemental material should appear at the end of the Materials and methods section.

Please note that JCB now requires authors to submit Source Data used to generate figures containing gels and Western blots with all revised manuscripts. This Source Data consists of fully uncropped and unprocessed images for each gel/blot displayed in the main and supplemental figures. Since your paper includes cropped gel and/or blot images, please be sure to provide one Source Data file for each figure that contains gels and/or blots along with your revised manuscript files. File names for Source Data figures should be alphanumeric without any spaces or special characters (i.e., SourceDataF#, where F# refers to the associated main figure number or SourceDataFS# for those associated with Supplementary figures). The lanes of the gels/blots should be labeled as they are in the associated figure, the place where cropping was applied should be marked (with a box), and molecular weight/size standards should be labeled wherever possible. Source Data files will be made available to reviewers during evaluation of revised manuscripts and, if your paper is eventually published in JCB, the files will be directly linked to specific figures in the published article.

The typical timeframe for revisions is three to four months. While most universities and institutes have reopened labs and allowed researchers to begin working at nearly pre-pandemic levels, we at JCB realize that the lingering effects of the COVID-19 pandemic may still be impacting some aspects of your work, including the acquisition of equipment and reagents. Therefore, if you anticipate any difficulties in meeting this aforementioned revision time limit, please contact us and we can work with you to

find an appropriate time frame for resubmission. Please note that papers are generally considered through only one revision cycle, so any revised manuscript will likely be either accepted or rejected.

Thank you for this interesting contribution to Journal of Cell Biology. You can contact us at the journal office with any questions at cellbio@rockefeller.edu.

Sincerely,

Michael Rout, PhD
Monitoring Editor
Journal of Cell Biology

Dan Simon, PhD
Scientific Editor
Journal of Cell Biology

Reviewer #1 (Comments to the Authors (Required)):

SUMMARY

In an interesting and well-prepared manuscript, Xu et al. report that NPAT, a nuclear protein essential for the formation of HLBs, binds to KPNA3 (rather than KPNA1, 2, 4, 5, 6) through the NLS sequence in the C-terminal region of NPAT. This interaction not only determines the nuclear localization of NPAT and formation of HLBs, but also prevents the self-association of NPAT in the cytoplasm. The authors also determined that there is a C-terminal self-interaction facilitator (C-SIF) motif localized next to the NLS that binds to the middle (M) region of NPAT and drives its self-association. Because of the proximity between C-SIF and NLS, the anchoring of KPNA3 to the NLS region sterically blocks the self-association of NPAT. The novelty of this work lies in the mechanisms by which i) NPAT is imported to the nucleus, ii) NPAT self-associates, and iii) KPNA3 modulates the self-association of NPAT. While the work is interesting and of good quality, there are a few points that need to be addressed to prove the claims summarised above.

Major concerns

- 1) In general, the most challenging aspect of this study is to separate the biology that is a direct consequence of mis localization due to impaired import from the effects of KPNA3 binding on the condensation of NPAT. The claim in this paper is that NPAT does not condense in the cytosol because it is bound to KPNA3, and it does condensate in the nucleus because KPNA3 is released. Figure 5I is important in this context, as the authors transfected the NPAT-deltaC-SIF construct and showed that while it can localize to the nucleus, it does not form condensates because it cannot self-associate. This corroborates the authors' claim that the formation of HLBs is condensation-dependent rather than simply localization-dependent. Also, the in vitro data support that condensation behavior is KPNA3-dependent, but there is no direct evidence for this in vivo. Experiments that would really be needed to prove the specific effects of KPNA3 on condensation in vivo are not presented. Three approaches come to my mind, but there may be more or better ones. A) to generate a mutant form of NPAT lacking the KPNA3-NLS while encoding an extra NLS for a different unrelated importin to ensure it is still imported. This mutant should show condensates in the cytosol as well as in the nucleus where it should still regulate RD-histones expression.
 - b) An approach to substantiate the main point of the paper would be to show that in the case of co-overexpression of PNAT and KPNA3 there is no RD-histones gene regulation, and to show that in IPs from nuclear fractions of these overexpressing cells, PNAT and KPNA3 are complexed. As a control, in a situation of wild type KPNA3 and PNAT levels, they should not be complexed in nuclear fractions.
 - c) a gene-fusion of PNAT-KPNA3 should show nuclear localization, but no condensation, and no RD-histones regulation.
- 2) The combined treatment of KPNA3 siRNA + ivermectin leads to cytoplasmic structures of NPAT, while the individual treatments only lead to reduced levels of NPAT in the nucleus, as well as less nuclear foci. The authors then conclude "This finding indicates that KPNA not only imports NPAT into nuclei, but also prevents aberrant NPAT condensation in cytoplasm". In order to confirm the specificity of KPNA3 in preventing aberrant NPAT condensation in the cytoplasm, a more straightforward approach would be to express the full-length NPAT without the NLS, which does not interact with KPNA3 and should form cytoplasmic structures. The authors should also include a discussion why the combined intervention of knockdown and ivermectin is needed.
- 3) In figure 4, the FLAG staining pattern in panel C is quite different compared to panel E. The authors should discuss to the least if and how the overexpression of NPAT influences the localization of KPNA3 in the nucleus?
 - 4) The knockdown of KPNA3 impairs the nuclear import of NPAT, which impairs the formation of HLBs and affects the

- expression levels of RD-histones. On the other hand, the overexpression of KPNA3 leads to a diffuse nuclear localization of NPAT. The authors should test if the overexpression of KPNA3 also affects the expression levels of RD-histones?
- 5) The statement "Thus, KPNA protein-dependent de-condensation of nuclear proteins could act as a general mechanism for regulating the formation of nonmembrane organelles in nuclei", in the discussion is a stretch. The word "general" is not justified.
 - 6) It would be helpful to show colocalization of KPNA3 and NPAT in the in vitro assays using partially labeled proteins.

Minor concerns

- 1) I find the discussion of putative possibilities for interventions too speculative at this point and hence not very valuable. I would recommend to remove that or tone it down.
- 2) The immunofluorescence images should show the separate channels.
- 3) Scale bar is missing from the insets in 3B
- 3) The Hoechst staining in figure 1F is strange. Any idea what is going on?
- 4) Could the authors explain the choice of the different cell lines 293T and MCF7?
- 5) I appreciate that the authors tested KNA1 through 6 and would suggest they make specific that they did exactly that, instead of using the more unclear phrasing of 'unbiased screen' which suggests an even broader screen than only the KPNA members.
- 6) In the in vitro experiments, please clarify if KPNA3 was added only after formation of particles or that the two were mixed from the start.
- 7) The phase separation assays are performed in the presence of PEG. Do the authors know if PEG is truly acting as a crowding agent or is it partitioning in the condensate? Fluorescent labeled PEG versions are available for this.
- 8) 5K is illustrative, as are the other models, but I would suggest adapting the NPAT condensate cartoon to a less organized, symmetrical form. The current cartoon suggests many things for which there is no evidence (and that are currently not relevant), such as the N-termini orienting towards the surface of the condensate."
- 9) The pH of the storage buffer is missing (p20)

Reviewer #2 (Comments to the Authors (Required)):

The manuscript by Xu et al uncovers a novel role for karyopherin alpha 3 as a regulator of histone locus body formation mediated by the scaffold protein NPAT, which undergoes phase separation in the nucleus. The authors show that NPAT is imported into the nucleus by karyopherin alpha 3 and that karyopherin alpha 3 also acts as a chaperone that prevents aberrant phase separation of NPAT in the cytoplasm. The authors then dissected the interacting regions involved in NPAT self-association as well as in NPAT-karyopherin alpha 3 interactions using biochemical and cell biological approaches. The results then led to a steric hindrance model/mechanism of karyopherin-mediated inhibition of NPAT phase separation in the cytoplasm.

Overall, these are important and novel findings, and the results are convincing. However, there are some points that should be addressed:

1. The manuscript would benefit from additional editing.
2. Fig. 1B should show an mRNA that does not change in levels as control, otherwise one may think that cells are dying under these conditions. The same number of cells should be compared under these conditions as cell proliferation may be decreased.
3. Figure 2A. NPAT blot is not clear. siRNA against NPAT should be used to demonstrate that those are indeed NPAT bands.
4. The graph in Figure 2B is partially cropped.
5. Graphs depicting nuclear or cytoplasmic GFP concentration: since comparisons were performed at a semiquantitative level and do not represent actual concentrations, it might be more appropriate to use "relative protein level" instead of protein concentration in the y-axis labels.
6. Figure 2C should be properly labeled: GFP is fused to NPAT (GFP-NPAT) but the label is showing GFP alone.
7. As in point #2, Figure 2F should also show an mRNA whose level is not altered as control, otherwise one may think that cells are dying under these conditions. Again, the same number of cells should be compared under these conditions as cell proliferation is decreased.
8. Figure 4B needs quantification to support the quantitative statement that NPAT self-association is decreased in the presence of KPNA3.
9. Define RD histone when first mentioned on page 3 line 62.
10. Line 228 "self-interaction of NAP. KPNA3 interacts with the NLC-containing 1031-1427C." I think NLC should be changed

to NLS.

Reviewer #3 (Comments to the Authors (Required)):

NPAT is a large, intrinsically disordered protein required for the assembly of histone locus bodies, which are found at replication dependent (RD) histone genes and are important for the cell cycle-regulated production of RD histone mRNAs. As properly regulated RD histone gene expression is critical for genome stability, and HLBs provide a tractable nuclear body to understand the mechanism of biomolecular condensate formation and function, understanding the regulation of NPAT function is important and should appeal broadly to the cell biological community. Here Xu et al. provide a thorough and well-presented study on the mechanisms of both NPAT nuclear import and phase separation properties. They make four main conclusions from their study:

1. The armadillo repeats 3-5 of karyopherin KPNA3 bind to an NLS in near the C-terminus of NPAT to translocate NPAT into the nucleus, thereby promoting HLB formation. The molecular and cell biology data supporting this conclusion are excellent and convincing, and represent a novel finding.
2. NPAT undergoes phase separation in vitro and GFP-NPAT displays properties of liquid-liquid phase separation (LLPS) in vivo. The experiments are convincing. The in vitro work is novel, but the in vivo work was demonstrated previously for the *Drosophila* NPAT homolog, Mxc. Nonetheless, demonstrating these properties of human NPAT is an important advance and necessary for our full understanding of HLB assembly in mammalian organisms.
3. KPNA3 prevents LLPS of NPAT both in vitro and in vivo by binding the NPAT NLS and disrupting an interaction between the middle, intrinsically disordered region of NPAT (i.e. amino acids 431-1030) and a newly identified self-interacting domain near the NPAT C-terminus (amino acids 1161-1290). This is an interesting observation that is well supported by the data.
4. KPNA3 binding of NPAT prevents NPAT from undergoing LLPS in the cytoplasm. This is the most interesting and novel conclusion from the work, but unfortunately seems to be only supported by one experiment: the appearance of NPAT foci in the cytoplasm by combined treatment of cells with si RNAs to KPNA3 and ivermectin. Given that ivermectin affects nuclear import more generally than just disruption of KPNA3, this single piece of data is not sufficient to make such a general conclusion, and represents the main criticism of the current manuscript.

Other issues that should be addressed by the authors, some of which pertain to the comments above.

1. The decrease in endogenous nuclear NPAT after KPNA3 depletion is only moderately convincing. It would help if these data could be strengthened, perhaps with a better KPNA3 knockdown.
2. That the combined KPNA3 KD and ivermectin treatment results in cytoplasm NPAT foci is quite interesting, but not fully fleshed out. Neither KPNA3 or ivermectin alone have this phenotype, and given that the KPNA3 KD is not complete this result suggests that ivermectin treatment is also "hypomorphic". Is it known that ivermectin treatment at the concentrations utilized in this study doesn't fully block KPNA3 function? What other karyopherins might be affected and thus potentially contribute to this phenotype?
3. Line 161: RD histone mRNA measurements always decrease when cell proliferation is attenuated, as occurs with KPNA3 KD. Consequently, the measured effect could entirely be due to the cell proliferation defect rather than the direct action of KPNA3. Thus, the authors need to be careful with their language and conclusion regarding whether KPNA3 function directly affects histone RD production.
4. Can the droplet size in the in vitro phase separation be quantified? The effects seem obvious but some kind of quantitative comparison would be more rigorous and perhaps helpful for interpreting the data.
5. The authors should express the KPNA3 delta ARM3-5 in vivo as in Figure 4E and ask if the NPAT foci are disrupted. That was convincing experiment.
6. Line 240: careful with language, as the data don't support "largely lost its self-association ability"; it's more attenuated and not really "lost".
7. Seems that all the effects in vivo result in a reduced number of normal looking NPAT foci. Thus the effect seems to be one of probability or frequency, which could be a result of reduced concentration of NPAT in the nucleus. The authors should consider this aspect of their data with respect to current ideas of biomolecular condensate formation.
8. The last section of the results regarding testing the steric hindrance model is unclear as written and should be reworked. Seems like the myc tag doesn't disrupt the interaction because it's too small, whereas the GFP tag is larger and then does. That

conception is not as clearly presented to the reader as it could be.

Minor suggestions and corrections.

Line 83: drosophila should be Drosophila

Throughout the manuscript the RD histone gene PCR is not measuring transcription but rather steady state accumulation of RD histone mRNA.

Line 133: perhaps mention also that the GFP-NPAT-C-region also doesn't form foci when accumulating in the nucleus, which actually fits the model that the SIF domain needs to interact with the M-region for LLPS and HLB formation.

The reduction of the number of NPAT foci after KPNA3 knockdown and other treatments is presumed to be a result of reduced nuclear concentration of NPAT, but other mechanisms could be involved and should probably be considered. For instance, does KPNA3 have other cargo that could possibly affect how NPAT forms foci in the nucleus?

Line 174: the GFP-NPAT structures don't look very spherical as stated, and thus don't appear to be typical LLPS. Also line 207 the structures don't look very "punctate".

Line 222: KNPA should be KPNA

In the discussion there are two references to Koreski et al 2020 that are incorrect. The correct citation is Terzo 2015, and the text should refer to Drosophila Mxc, not NPAT.

A point-by-point response to the reviewers (Submission of revised JCB manuscript #202401036R):

Reviewer #1

Summary

In an interesting and well-prepared manuscript, Xu et al. report that NPAT, a nuclear protein essential for the formation of HLBs, binds to KPNA3 (rather than KPNA1, 2, 4, 5, 6) through the NLS sequence in the C-terminal region of NPAT. This interaction not only determines the nuclear localization of NPAT and formation of HLBs, but also prevents the self-association of NPAT in the cytoplasm. The authors also determined that there is a C-terminal self-interaction facilitator (C-SIF) motif localized next to the NLS that binds to the middle (M) region of NPAT and drives its self-association. Because of the proximity between C-SIF and NLS, the anchoring of KPNA3 to the NLS region sterically blocks the self-association of NPAT.

The novelty of this work lies in the mechanisms by which i) NPAT is imported to the nucleus, ii) NPAT self-associates, and iii) KPNA3 modulates the self-association of NPAT. While the work is interesting and of good quality, there are a few points that need to be addressed to prove the claims summarised above.

Response:

We would like to express our gratitude for your positive feedback and insightful comments. Accordingly, we have made revisions to the paper, which are reflected below.

Major concerns

1) In general, the most challenging aspect of this study is to separate the biology that is a direct consequence of mislocalization due to impaired import from the effects of KPNA3 binding on the condensation of NPAT. The claim in this paper is that NPAT does not condense in the cytosol because it is bound to KPNA3, and it does condensate in the nucleus because KPNA3 is released. Figure 5I is important in this context, as the authors transfected the NPAT-deltaC-SIF construct and showed that while it can localize to the nucleus, it does not form condensates because it cannot self-associate. This

corroborates the authors' claim that the formation of HLBs is condensation-dependent rather than simply localization-dependent. Also, the *in vitro* data support that condensation behavior is KPNA3-dependent, but there is no direct evidence for this *in vivo*. Experiments that would really be needed to prove the specific effects of KPNA3 on condensation *in vivo* are not presented.

Three approaches come to my mind, but there may be more or better ones.

a) to generate a mutant form of NPAT lacking the KPNA3-NLS while encoding an extra NLS for a different unrelated importin to ensure it is still imported. This mutant should show condensates in the cytosol as well as in the nucleus where it should still regulate RD-histones expression.

b) An approach to substantiate the main point of the paper would be to show that in the case of co-overexpression of PNAT and KPNA3 there is no RD-histones gene regulation, and to show that in IPs from nuclear fractions of these overexpressing cells, PNAT and KPNA3 are complexed. As a control, in a situation of wild type KPNA3 and PNAT levels, they should not be complexed in nuclear fractions.

c) a gene-fusion of PNAT-KPNA3 should show nuclear localization, but no condensation, and no RD-histones regulation.

Response:

We fully agree that it is important to distinguish the effects of mislocalization due to impaired import from the impact of KPNA3 binding on NPAT condensation. We also thank you for suggesting the experiments. Replacing the NPAT-NLS with other NLS is a good idea, though we are concerned that other NLS could attract different KPNA3s, which might also sterically block the self-association of NPAT. Inspired by your idea, we performed an alternative experiment to address the issue. We swap the ARM 3-5 of KPNA5, which does not interact with NPAT (Fig. 1D) and could not suppress HLB formation (new Fig. 5A), with the ARM 3-5 of KPNA3 (new Fig. 5B). Interestingly, the KPNA5-3-hybrid mutant, which contains the KPNA3 ARM 3-5 region (new Fig. 5B), can bind the NLS region of NPAT (new Fig. 5C) and subsequently sterically block the formation of HLB (new Fig. 5D).

Moreover, we also prepared an NPAT-KPNA3 fusion protein as you instructed

(new Fig. 5E and 5F). This mutant is to mimic the permanent interaction between NPAT and KPNA3. Notably, the immunofluorescence study showed that this mutant displayed nuclear localization, but no condensate formation (new Fig. 5G). These findings further solidify the conclusion that binding with KPNA3 suppresses NPAT condensation.

2) The combined treatment of KPNA3 siRNA + ivermectin leads to cytoplasmic structures of NPAT, while the individual treatments only lead to reduced levels of NPAT in the nucleus, as well as fewer nuclear foci. The authors then conclude "This finding indicates that KPNA not only imports NPAT into nuclei but also prevents aberrant NPAT condensation in cytoplasm". To confirm the specificity of KPNA3 in preventing aberrant NPAT condensation in the cytoplasm, a more straightforward approach would be to express the full-length NPAT without the NLS, which does not interact with KPNA3 and should form cytoplasmic structures.

The authors should also include a discussion of why the combined intervention of knockdown and ivermectin is needed.

Response:

We agree that it is important to confirm the specificity of KPNA3 in suppressing aberrant NPAT condensation. We also appreciate your suggestion to use a more straightforward approach that examines the NPAT- Δ NLS mutant in cells. This mutant was created accordingly (new Fig. 4L). Indeed, the NPAT- Δ NLS mutant displayed cytoplasmic foci as you postulated (new Fig. 4M).

We also thank you for pointing out that we should explain the combined usage of KPNA siRNA and ivermectin. Aberrant cytoplasmic NPAT foci formation could be observed in cells treated with a higher concentration of ivermectin (50 μ M) (new Fig. S2C) or when we extended the treatment time of KPNA3 siRNA (from 48 hours to 72 hours) (new Fig. S2E). However, a combination of both interventions led to more significant effects (revised Fig. 2E). We thus included the data of two single treatments in the text.

3) In Figure 4, the FLAG staining pattern in panel C is quite different compared to panel E. The authors should discuss to the least if and how the overexpression of

NPAT influences the localization of KPNA3 in the nucleus.

Response:

We thank you for raising the point. We thus discussed this issue in the revised discussion section.

4) The knockdown of KPNA3 impairs the nuclear import of NPAT, which impairs the formation of HLBs and affects the expression levels of RD-histones. On the other hand, the overexpression of KPNA3 leads to a diffuse nuclear localization of NPAT. The authors should test if the overexpression of KPNA3 also affects the expression levels of RD-histones.

Response:

We thank the reviewer for suggesting the experiment. We performed qPCR analysis and found that overexpression of KPNA3 reduced the expression levels of RD-histones (new Fig. 4F).

5) The statement "Thus, KPNA protein-dependent de-condensation of nuclear proteins could act as a general mechanism for regulating the formation of nonmembrane organelles in nuclei", in the discussion is a stretch. The word "general" is not justified.

Response:

We agree that it has not been justified to claim a general mechanism. We thus revised the sentence accordingly.

6) It would be helpful to show colocalization of KPNA3 and NPAT in the in vitro assays using partially labeled proteins.

Response:

We thank you for suggesting the experiment and will perform it in future studies.

Minor concerns

1) I find the discussion of putative possibilities for interventions too speculative at this point and hence not very valuable. I would recommend to remove that or tone it down.

Response:

We agree that it was too speculative at the intervention point without experimental evidence. We thus revised the discussion section to tone down it.

2) The immunofluorescence images should show the separate channels.

Response:

We displayed separate channels as shown in revised Fig. 1A, revised Fig. 1C, revised Fig. 1H, revised Fig. 2D, revised Fig. 2E, new Fig. S2C, new Fig. S2E, new Fig. 5A, new Fig. 5D, new Fig. 5G, and revised Fig. 6I.

3) Scale bar is missing from the insets in 3B

Response:

We added a Scale bar in the revised Fig. 3B.

4) The Hoechst staining in figure 1F is strange. Any idea what is going on?

Response:

Fig. 1F is co-immunoprecipitation and Fig. 1H is immunofluorescence staining. We repeated the experiments of Fig. 1H and provided a new image (revised Fig. 1H).

5) Could the authors explain the choice of the different cell lines 293T and MCF7?

Response:

We agree that the choice of cell lines needs to be explained. 293T cell line was used for co-immunoprecipitation assay because of its ease of transfection and production of recombinant proteins (Stepanenko and Dmitrenko, 2015). MCF-7 breast cancer cell line was used for immunofluorescence study because of its ease of transfection and the association of NPAT with breast cancer (Milne et al., 2017; Rogers et al., 2015). This information was included in the text.

6) I appreciate that the authors tested KPNA1 through 6 and would suggest they make specific that they did exactly that, instead of using the more unclear phrasing of 'unbiased screen' which suggests an even broader screen than only the KPNA members.

Response:

We appreciate your instruction and have removed “using unbiased screening” from the abstract and the discussion section.

7) In the in vitro experiments, please clarify if KPNA3 was added only after formation of particles or that the two were mixed from the start.

Response:

We agree that this information needs to be included and revised in the material and methods section to demonstrate that the two proteins were mixed from the start.

8) The phase separation assays are performed in the presence of PEG. Do the authors know if PEG is truly acting as a crowding agent or is it partitioning in the condensate? Fluorescent labeled PEG versions are available for this.

Response:

We believe you have raised a fundamental question in the field of phase separation. The commonly used PEG for in vitro condensation assay is considered an inert crowding agent. However, recent studies revealed that PEG might affect the phase separation of some proteins (Alfano et al., 2024; Andre et al., 2023; Park et al., 2020). We plan to address this important issue in our future study systematically.

9) 5K is illustrative, as are the other models, but I would suggest adapting the NPAT condensate cartoon to a less organized, symmetrical form. The current cartoon suggests many things for which there is no evidence (and that are currently not relevant), such as the N-termini orienting towards the surface of the condensate."

Response:

We agree that the cartoon is not fully supported by the data and revised it according to your instructions (revised Fig. 6K).

10) The pH of the storage buffer is missing (p20)

Response:

We thank you for pointing out this issue and revised the text accordingly.

Reviewer #2**Comments to the Authors:**

The manuscript by Xu et al uncovers a novel role for karyopherin alpha 3 as a regulator of histone locus body formation mediated by the scaffold protein NPAT, which undergoes phase separation in the nucleus. The authors show that NPAT is imported into the nucleus by karyopherin alpha 3 and that karyopherin alpha 3 also acts as a chaperone that prevents aberrant phase separation of NPAT in the cytoplasm. The authors then dissected the interacting regions involved in NPAT self-association as well as in NPAT-karyopherin alpha 3 interactions using biochemical and cell biological approaches. The results then led to a steric hindrance model/mechanism of karyopherin-mediated inhibition of NPAT phase separation in the cytoplasm.

Overall, these are important and novel findings, and the results are convincing. However, there are some points that should be addressed:

Response:

We sincerely thank you for the positive comments on our manuscript. We are grateful for the constructive criticisms which we have addressed to our best.

1. The manuscript would benefit from additional editing.

Response:

We have revised the manuscript to improve it further.

2. Fig. 1B should show an mRNA that does not change in levels as control, otherwise one may think that cells are dying under these conditions. The same number of cells should be compared under these conditions as cell proliferation may be

decreased.

Response:

We thank you for suggesting the experiments. Therefore, we examined the expression levels of Hexokinase 1 (HK1) and Enolase 1 (ENO1) as controls and used the same number of cells in different groups (revised Fig. 1B, revised Fig. 2F, new Fig. 4F).

3. Figure 2A. NPAT blot is not clear. siRNA against NPAT should be used to demonstrate that those are indeed NPAT bands.

Response:

We agree that the NPAT band needs to be verified. As instructed, we validated the NPAT band by using siRNA-mediated NPAT knockdown as shown in new Fig. S2A.

4. The graph in Figure 2B is partially cropped.

Response:

We thank you for pointing out this issue and revised the graph accordingly (revised Fig. 2B).

5. Graphs depicting nuclear or cytoplasmic GFP concentration: since comparisons were performed at a semiquantitative level and do not represent actual concentrations, it might be more appropriate to use "relative protein level" instead of protein concentration in the y-axis labels.

Response:

We thank you for raising this point and revised the Figures accordingly (revised

Fig. 2A, revised Fig. 2C, revised Fig.4D, revised Fig. 6J, revised Fig. S2B).

6. Figure 2C should be properly labeled: GFP is fused to NPAT (GFP-NPAT) but the label is showing GFP alone.

Response:

We thank you for pointing out this mistake and revised the label of y-axis accordingly (revised Fig. 2C).

7. As in point #2, Figure 2F should also show an mRNA whose level is not altered as control, otherwise one may think that cells are dying under these conditions. Again, the same number of cells should be compared under these conditions as cell proliferation is decreased.

Response:

We agree that appropriate control needs to be included to ensure that cells are not dying. We performed the assay again accordingly (revised Fig. 2F).

8. Figure 4B needs quantification to support the quantitative statement that NPAT self-association is decreased in the presence of KPNA3.

Response:

We agree that quantitative statements need to be supported by solid evidence. The quantification analysis was provided as shown in revised Fig. 4B.

9. Define RD histone when first mentioned on page 3 line 62.

Response:

We thank you for raising this point and revised the text accordingly.

10. Line 228 "self-interaction of NPAT. KPNA3 interacts with the NLC-containing 1031-1427C." I think NLC should be changed to NLS.

Response:

We thank you for pointing out this typo and have revised it accordingly.

Reviewer #3

Comments to the Authors:

NPAT is a large, intrinsically disordered protein required for the assembly of histone locus bodies, which are found at replication dependent (RD) histone genes and are important for the cell cycle-regulated production of RD histone mRNAs. As properly regulated RD histone gene expression is critical for genome stability, and HLBs provide a tractable nuclear body to understand the mechanism of biomolecular condensate formation and function, understanding the regulation of NPAT function is important and should appeal broadly to the cell biological community. Here Xu et al. provide a thorough and well-presented study on the mechanisms of both NPAT nuclear import and phase separation properties. They make four main conclusions from their study:

1. The armadillo repeats 3-5 of karyopherin KPNA3 bind to an NLS in near the C-terminus of NPAT to translocate NPAT into the nucleus, thereby promoting HLB formation. The molecular and cell biology data supporting this conclusion are excellent and convincing, and represent a novel finding.

2. NPAT undergoes phase separation in vitro and GFP-NPAT displays properties

of liquid-liquid phase separation (LLPS) in vivo. The experiments are convincing. The in vitro work is novel, but the in vivo work was demonstrated previously for the *Drosophila* NPAT homolog, Mxc. Nonetheless, demonstrating these properties of human NPAT is an important advance and necessary for our full understanding of HLB assembly in mammalian organisms.

3. KPNA3 prevents LLPS of NPAT both in vitro and in vivo by binding the NPAT NLS and disrupting an interaction between the middle, intrinsically disordered region of NPAT (i.e. amino acids 431-1030) and a newly identified self-interacting domain near the NPAT C-terminus (amino acids 1161-1290). This is an interesting observation that is well supported by the data.

4. KPNA3 binding of NPAT prevents NPAT from undergoing LLPS in the cytoplasm. This is the most interesting and novel conclusion from the work, but unfortunately seems to be only supported by one experiment: the appearance of NPAT foci in the cytoplasm by combined treatment of cells with si RNAs to KPNA3 and ivermectin. Given that ivermectin affects nuclear import more generally than just disruption of KPNA3, this single piece of data is not sufficient to make such a general conclusion, and represents the main criticism of the current manuscript.

Response:

We sincerely appreciate your positive comments and agree that more evidence must be presented to validate our model that KPNA3 binding of NPAT prevents NPAT from undergoing LLPS in the cytoplasm. To this aim, we used a straightforward approach by examining the NPAT- Δ NLS mutant (new Fig. 4L), which displays

cytoplasmic retention (new Fig. 4M). Indeed, the NPAT- Δ NLS mutant also displayed cytoplasmic foci as shown in new Fig. 4M.

Other issues that should be addressed by the authors, some of which pertain to the comments above.

1. The decrease in endogenous nuclear NPAT after KPNA3 depletion is only moderately convincing. It would help if these data could be strengthened, perhaps with a better KPNA3 knockdown.

Response:

We agree with the reviewer that data showing better KPNA3 knockdown should be presented. As instructed by the reviewer, we provided strengthened data as shown in revised Fig. 2A.

2. That the combined KPNA3 KD and ivermectin treatment results in cytoplasm NPAT foci is quite interesting, but not fully fleshed out. Neither KPNA3 or ivermectin alone have this phenotype, and given that the KPNA3 KD is not complete this result suggests that ivermectin treatment is also "hypomorphic". Is it known that ivermectin treatment at the concentrations utilized in this study doesn't fully block KPNA3 function? What other karyopherins might be affected and thus potentially contribute to this phenotype?

Response:

We thank you for raising this issue. Either elevating ivermectin concentration (from 25 μ M to 50 μ M) or extending the siRNA knockdown time (from 48 hours to 72

hours) could result in cytoplasmic foci of NPAT (new Fig. S2C and new Fig. S2E), while a combination of siRNA-mediated knockdown and ivermectin treatment leads to more significant cytoplasmic NPAT foci formation (revised Fig. 2E). We thus included the data of ivermectin treatment and time-extended KPNA3 knockdown in the text to clarify this issue.

3. Line 161: RD histone mRNA measurements always decrease when cell proliferation is attenuated, as occurs with KPNA3 KD. Consequently, the measured effect could entirely be due to the cell proliferation defect rather than the direct action of KPNA3. Thus, the authors need to be careful with their language and conclusion regarding whether KPNA3 function directly affects histone RD production.

Response:

We agree that we should be more careful when describing our data. We thus revised the text accordingly.

4. Can the droplet size in the in vitro phase separation be quantified? The effects seem obvious but some kind of quantitative comparison would be more rigorous and perhaps helpful for interpreting the data.

Response:

We agree that quantification analysis for in vitro phase separation should be performed. As shown in revised Fig. 3D, revised Fig. 3E, revised Fig. 3F, revised Fig. 3G, revised Fig. 3H, revised Fig. 4A, revised Fig. 4K, revised Fig. S4, revised Fig. 6G, revised Fig. 7C, revised Fig. 7E, and revised Fig. 7H, we quantified the size of the

NPAT droplets.

5. The authors should express the KPNA3 delta ARM3-5 in vivo as in Figure 4E and ask if the NPAT foci are disrupted. That was convincing experiment.

Response:

We agree that it is of importance to verify the effects of ARM 3-5 in cells. Since the deletion of ARM 3-5 might potentially affect the 3-D structure of KPNA3, we performed an alternative experiment to address the issue. We swap the ARM 3-5 of KPNA5, which does not interact with NPAT (Fig. 1D) and could not suppress HLB formation (new Fig. 5A), with the ARM 3-5 of KPNA3 (new Fig. 5B). Interestingly, the mutant KPNA5 (KPNA5-3-hybrid mutant), which contains the KPNA3 ARM 3-5 region, can bind NPAT and subsequently sterically block the formation of HLB (new Fig. 5C, new Fig. 5D).

6. Line 240: careful with language, as the data don't support "largely lost its self-association ability"; it's more attenuated and not really "lost".

Response:

We thank you for pointing out this issue. We thus revised the sentence accordingly.

7. Seems that all the effects in vivo result in a reduced number of normal looking NPAT foci. Thus the effect seems to be one of probability or frequency, which could be a result of reduced concentration of NPAT in the nucleus. The authors should consider this aspect of their data with respect to current ideas of biomolecular condensate formation.

Response:

We agree that the concentration of proteins is involved in forming biomolecular condensation. As shown in Fig. 2A and 2C, the knockdown of KPNA3 reduced the expression levels of NPAT in the nucleus. The decreased concentration of nuclear NPAT resulted in reduced NPAT-positive foci in nuclei (Fig. 2B and 2D). We also included this issue in the discussion section.

8. The last section of the results regarding testing the steric hindrance model is unclear as written and should be reworked. Seems like the myc tag doesn't disrupt the interaction because it's too small, whereas the GFP tag is larger and then does. That conception is not as clearly presented to the reader as it could be.

Response:

We thank you for raising this point and reworded the last section of the results to clarify this important issue.

Minor suggestions and corrections.

1. Line 83: drosophila should be Drosophila

Response:

We thank you for pointing out this mistake.

2. Throughout the manuscript the RD histone gene PCR is not measuring transcription but rather steady state accumulation of RD histone mRNA.

Response:

We thank you for raising this point and revised the text to clarify that we measured the expression levels of RD histone mRNA, rather than the transcription of RD histones.

3. Line 133: perhaps mention also that the GFP-NPAT-C-region also doesn't form foci when accumulating in the nucleus, which actually fits the model that the SIF domain needs to interact with the M-region for LLPS and HLB formation.

Response:

We appreciate the thought and included this important point in the discussion section.

4. The reduction of the number of NPAT foci after KPNA3 knockdown and other treatments is presumed to be a result of reduced nuclear concentration of NPAT, but other mechanisms could be involved and should probably be considered. For instance, does KPNA3 have other cargo that could possibly affect how NPAT forms foci in the nucleus?

Response:

We agree with the reviewer that there remains the possibility that other cargoes of KPNA3 might indirectly contribute to the reduction of NPAT foci number. We discussed this issue in the revised discussion section.

5. Line 174: the GFP-NPAT structures don't look very spherical as stated, and thus don't appear to be typical LLPS. Also line 207 the structures don't look very "punctate".

Response:

We thank you for raising this issue and have performed sphericity analysis for NPAT droplets as shown in new Fig. S3A and new Fig. S3B.

6. Line 222: KNPA should be KPNA

Response:

We thank you for pointing out this typo and revised it accordingly.

7. In the discussion there are two references to Koreski et al 2020 that are incorrect. The correct citation is Terzo 2015, and the text should refer to *Drosophila* Mxc, not NPAT.

Response:

We thank you for pointing out this mistake and have revised it accordingly.

References

- Alfano, C., Y. Fichou, K. Huber, M. Weiss, E. Spruijt, S. Ebbinghaus, G. De Luca, M.A. Morando, V. Vetri, P.A. Temussi, and A. Pastore. 2024. Molecular Crowding: The History and Development of a Scientific Paradigm. *Chem Rev.* 124:3186-3219.
- Andre, A.A.M., N.A. Yewdall, and E. Spruijt. 2023. Crowding-induced phase separation and gelling by co-condensation of PEG in NPM1-rRNA condensates. *Biophys J.* 122:397-407.
- Milne, R.L., K.B. Kuchenbaecker, K. Michailidou, J. Beesley, S. Kar, S. Lindstrom, S. Hui, A. Lemaçon, P. Soucy, J. Dennis, X. Jiang, A. Rostamianfar, H. Finucane, M.K. Bolla, L. McGuffog, Q. Wang, C.M. Aalfs, M. Adams, J. Adlard, S. Agata, S. Ahmed, H. Ahsan, K. Aittomäki, F. Al-Ejeh, J. Allen, C.B. Ambrosone, C.I. Amos, I.L. Andrulis, H. Anton-Culver, N.N. Antonenkova, V. Arndt, N. Arnold, K.J. Aronson, B. Auber, P.L. Auer, M.G.E.M. Ausems, J. Azzollini, F. Bacot, J. Balmaña, M. Barile, L. Barjhoux, R.B. Barkardottir, M. Barrdahl, D. Barnes, D. Barrowdale, C. Baynes, M.W. Beckmann, J. Benitez, M. Bermisheva, L. Bernstein, Y.J. Bignon, K.R. Blazer, M.J. Blok, C. Blomqvist, W. Blot, K. Bobolis, B. Boeckx, N.V. Bogdanova, A. Bojesen, S.E. Bojesen, B. Bonanni,

- A.L. Borresen-Dale, A. Bozsik, A.R. Bradbury, J.S. Brand, H. Brauch, H. Brenner, B. Bressac-de Paillerets, C. Brewer, L. Brinton, P. Broberg, A. Brooks-Wilson, J. Brunet, T. Brüning, B. Burwinkel, S.S. Buys, J.Y. Byun, Q.Y. Cai, T. Caldés, M.A. Caligo, I. Campbell, F. Canzian, O. Caron, A. Carracedo, B.D. Carter, J.E. Castelao, L. Castera, V. Caux-Moncoutier, S.B. Chan, J. Chang-Claude, S.J. Chanock, X.Q. Chen, T.Y.D. Cheng, J. Chiquette, H. Christiansen, K.B.M. Claes, C.L. Clarke, T. Conner, D.M. Conroy, J. Cook, et al. 2017. Identification of ten variants associated with risk of estrogen-receptor-negative breast cancer. *Nat Genet.* 49:1767-1778.
- Park, S., R. Barnes, Y. Lin, B.J. Jeon, S. Najafi, K.T. Delaney, G.H. Fredrickson, J.E. Shea, D.S. Hwang, and S. Han. 2020. Dehydration entropy drives liquid-liquid phase separation by molecular crowding. *Commun Chem.* 3:83.
- Rogers, S., B.S. Gloss, C.S. Lee, C.M. Sergio, M.E. Dinger, E.A. Musgrove, A. Burgess, and C.E. Caldon. 2015. Cyclin E2 is the predominant E-cyclin associated with NPAT in breast cancer cells. *Cell Div.* 10:1.
- Stepanenko, A.A., and V.V. Dmitrenko. 2015. HEK293 in cell biology and cancer research: phenotype, karyotype, tumorigenicity, and stress-induced genome-phenotype evolution. *Gene.* 569:182-190.

August 28, 2024

RE: JCB Manuscript #202401036R

Dr. Li Ling Zheng
Second Affiliated Hospital of Zhejiang University

Dear Dr. Zheng,

Thank you for submitting your revised manuscript entitled "Nuclear import adaptor KPNA3 regulates histone locus body formation by modulating the condensation and nuclear import of NPAT." We would be happy to publish your paper in JCB pending final revisions necessary to address the remaining minor reviewer comment and meet our formatting guidelines (see details below).

A. MANUSCRIPT ORGANIZATION AND FORMATTING:

1) Text limits: Character count for Articles is < 40,000, not including spaces. Count includes title page, abstract, introduction, results, discussion, and acknowledgments. Count does not include materials and methods, figure legends, references, tables, or supplemental legends.

2) Figure formatting: Articles may have up to 10 main text figures. Scale bars must be present on all microscopy images, including inset magnifications. Molecular weight or nucleic acid size markers must be included on all gel electrophoresis. Please add scale bars for inset magnifications in Figures 1A/C, 2B/D/E, 4C/E, 5A/D, 6I, S1B/C, & S2C/E.

Also, please avoid pairing red and green for images and graphs to ensure legibility for color-blind readers. If red and green are paired for images, please ensure that the particular red and green hues used in micrographs are distinctive with any of the colorblind types. If not, please modify colors accordingly or provide separate images of the individual channels.

3) Statistical analysis: Error bars on graphic representations of numerical data must be clearly described in the figure legend. The number of independent data points (n) represented in a graph must be indicated in the legend. Please, indicate whether 'n' refers to technical or biological replicates (i.e. number of analyzed cells, samples or animals, number of independent experiments). If independent experiments with multiple biological replicates have been performed, we recommend using distribution-reproducibility SuperPlots (please see Lord et al., JCB 2020) to better display the distribution of the entire dataset, and report statistics (such as means, error bars, and P values) that address the reproducibility of the findings.

Statistical methods should be explained in full in the materials and methods. For figures presenting pooled data the statistical measure should be defined in the figure legends. Please also be sure to indicate the statistical tests used in each of your experiments (both in the figure legend itself and in a separate methods section) as well as the parameters of the test (for example, if you ran a t-test, please indicate if it was one- or two-sided, etc.). Also, if you used parametric tests, please indicate if the data distribution was tested for normality (and if so, how). If not, you must state something to the effect that "Data distribution was assumed to be normal but this was not formally tested."

4) Abstract and title: Please revise the abstract as suggested by Reviewer #1. The title should be less than 100 characters including spaces. Your current title is over the limit so we suggest shortening it to:

"KPNA3 regulates histone locus body formation by modulating condensation and nuclear import of NPAT"

5) Materials and methods: Should be comprehensive and not simply reference a previous publication for details on how an experiment was performed. Please provide full descriptions (at least in brief) in the text for readers who may not have access to referenced manuscripts. The text should not refer to methods "...as previously described." Please also indicate the type of membrane used for immunoblotting.

6) For all cell lines, vectors, constructs/cDNAs, etc. - all genetic material: please include database / vendor ID (e.g., Addgene, ATCC, etc.) or if unavailable, please briefly describe their basic genetic features, even if described in other published work or gifted to you by other investigators (and provide references where appropriate). Please be sure to provide the sequences for all of your oligos: primers, si/shRNA, RNAi, gRNAs, etc. in the materials and methods. You must also indicate in the methods the

source, species, and catalog numbers/vendor identifiers (where appropriate) for all of your antibodies, including secondary. If antibodies are not commercial, please add a reference citation if possible.

7) Microscope image acquisition: The following information must be provided about the acquisition and processing of images:

- a. Make and model of microscope
- b. Type, magnification, and numerical aperture of the objective lenses
- c. Temperature
- d. Imaging medium
- e. Fluorochromes
- f. Camera make and model
- g. Acquisition software
- h. Any software used for image processing subsequent to data acquisition. Please include details and types of operations involved (e.g., type of deconvolution, 3D reconstitutions, surface or volume rendering, gamma adjustments, etc.).

8) References: There is no limit to the number of references cited in a manuscript. References should be cited parenthetically in the text by author and year of publication. Abbreviate the names of journals according to PubMed.

9) Supplemental materials: Articles may have up to 5 supplemental figures and 10 videos. Please also note that tables, like figures, should be provided as individual, editable files. A summary of all supplemental material should appear at the end of the Materials and methods section. Please include one brief sentence per item.

10) eTOC summary: A ~40-50 word summary that describes the context and significance of the findings for a general readership should be included on the title page. The statement should be written in the present tense and refer to the work in the third person. It should begin with "First author name(s) et al..." to match our preferred style.

11) Conflict of interest statement: JCB requires inclusion of a statement in the acknowledgements regarding competing financial interests. If no competing financial interests exist, please include the following statement: "The authors declare no competing financial interests." If competing interests are declared, please follow your statement of these competing interests with the following statement: "The authors declare no further competing financial interests."

12) A separate author contribution section is required following the Acknowledgments in all research manuscripts. All authors should be mentioned and designated by their first and middle initials and full surnames. We encourage use of the CRediT nomenclature (<https://casrai.org/credit/>).

13) ORCID IDs: ORCID IDs are unique identifiers allowing researchers to create a record of their various scholarly contributions in a single place. Please note that ORCID IDs are required for all authors. At resubmission of your final files, please be sure to provide your ORCID ID and those of all co-authors.

14) JCB requires authors to submit Source Data used to generate figures containing gels and Western blots with all revised manuscripts. This Source Data consists of fully uncropped and unprocessed images for each gel/blot displayed in the main and supplemental figures. Since your paper includes cropped gel and/or blot images, please be sure to provide one Source Data file for each figure that contains gels and/or blots along with your revised manuscript files. File names for Source Data figures should be alphanumeric without any spaces or special characters (i.e., SourceDataF#, where F# refers to the associated main figure number or SourceDataFS# for those associated with Supplementary figures). The lanes of the gels/blots should be labeled as they are in the associated figure, the place where cropping was applied should be marked (with a box), and molecular weight/size standards should be labeled wherever possible. Source Data files will be directly linked to specific figures in the published article.

15) Journal of Cell Biology now requires a data availability statement for all research article submissions. These statements will be published in the article directly above the Acknowledgments. The statement should address all data underlying the research presented in the manuscript. Please visit the JCB instructions for authors for guidelines and examples of statements at (<https://rupress.org/jcb/pages/editorial-policies#data-availability-statement>).

B. FINAL FILES:

Thank you for your attention to these final processing requirements. Please revise and format the manuscript and upload materials within 7 days. If you need an extension for whatever reason, please let us know and we can work with you to determine a suitable revision period.

Thank you for this interesting contribution, we look forward to publishing your paper in Journal of Cell Biology.

Sincerely,

Michael Rout, PhD
Monitoring Editor
Journal of Cell Biology

Dan Simon, PhD
Scientific Editor
Journal of Cell Biology

Reviewer #1 (Comments to the Authors (Required)):

I have read the revised version of the manuscript with much interest and am very happy to see that all the additional experiments performed further support and strengthen the original conclusions. Congratulations!

A minor suggestion is to change "anchoring" to "binding" in the newly added abstract sentence "We identify KPNA3 as a specific importin that drives the nuclear import of NPAT by anchoring to the nuclear localization signal (NLS) sequence"

Reviewer #3 (Comments to the Authors (Required)):

The authors improve an already good manuscript by adding experiments that further support the most novel and interesting aspect of the study: that KPNA3 binding to an NLS in NPAT prevents NPAT condensate formation in the cytoplasm. This observation, in addition to convincing evidence that KPNA3 chaperones NPAT into the nucleus and as such is important for HLB formation, make this a nice study that should appeal broadly.

Important typo line 239: "Firstly, we created a KPNA5-3-hybrid mutant by swapping the ARM 3-5 of KPNA with the ARM 3-5 of KPNA3 (Fig. 5B)." Need to specify KPNA5!